# Beyond Homogeneous Attention: Memory-Efficient LLMs via Fourier-Approximated KV Cache

## Abstract

Large Language Models struggle with memory demands from the growing Key-Value (KV) cache as context lengths increase. Existing compression methods homogenize head dimensions or rely on attention-guided token pruning, often sacrificing accuracy or introducing computational overhead. We propose **Fourier-Attention**, a training-free framework that exploits the heterogeneous roles of transformer head dimensions: lower dimensions prioritize local context, while upper ones capture long-range dependencies. By projecting the long-context-insensitive dimensions onto orthogonal Fourier bases, FourierAttention approximates their temporal evolution with fixed-length spectral coefficients. Evaluations on LLaMA models show FourierAttention achieves the best long-context accuracy on Long-Bench and Needle-In-A-Haystack (NIAH). Besides, a custom Triton kernel, **Flash-FourierAttention**, is designed to optimize memory via streamlined read-write operations, enabling efficient deployment without performance compromise.

## 1 Introduction

Large Language Models (LLMs) have transformed natural language processing with breakthroughs in text generation, comprehension, and reasoning (OpenAI, 2023; Sun et al., 2024; OpenAI, 2024; Guo et al., 2025). However, their autoregressive decoding relies heavily on a memory-intensive Key-Value (KV) cache, leading to significant memory allocation as context lengths scale (Vaswani et al., 2017; Fu, 2024; Liu et al., 2025). This overhead limits LLM deployment in resource-constrained environments. While approaches like sparse attention and cache compression have been explored to reduce memory needs, they often compromise accuracy or add complexity (Cai et al., 2024; Yuan et al., 2025). Developing memory-efficient methods that preserve performance remains crucial for the broader applicability of LLMs.

Existing training-free KV cache compression methods, like token eviction strategies (Xiao et al., 2024; Zhang et al., 2023; Li et al., 2024b), prune sequence subsets but overlook the heterogeneous roles of head dimensions, leaving dimension-aware allocation largely unexplored. Similarly, hidden dimension compression (Chang et al., 2024; Saxena et al., 2024) methods apply uniform ratios, both neglect their distinct contribution across dimensions (Liu et al., 2024b; Peng et al., 2024). These approaches treat head dimensions as homogeneous, static units rather than dynamically allocating resources based on their importance.

Another critical limitation of existing methods lies in their reliance on attention-guided strategies (Zhang et al., 2023; Li et al., 2024b). While these approaches enable selective token pruning with minimal accuracy degradation, they impose prohibitive memory and latency overheads due to attention score recalculation. We address this challenge by adapting the HiPPO framework (Gu et al., 2020), a mathematically grounded approach for long-sequence modeling. HiPPO approximates infinite-length sequences as compact finite states by projecting inputs onto finite-order orthogonal basis functions, such as polynomial bases or Fourier bases (Gu et al., 2020; He et al., 2023). This retains global critical and contextually vital patterns while filtering out redundant signals. By leveraging HiPPO's theoretical foundations, we can bypass attention recomputation entirely, achieving both memory efficiency and computational efficiency.

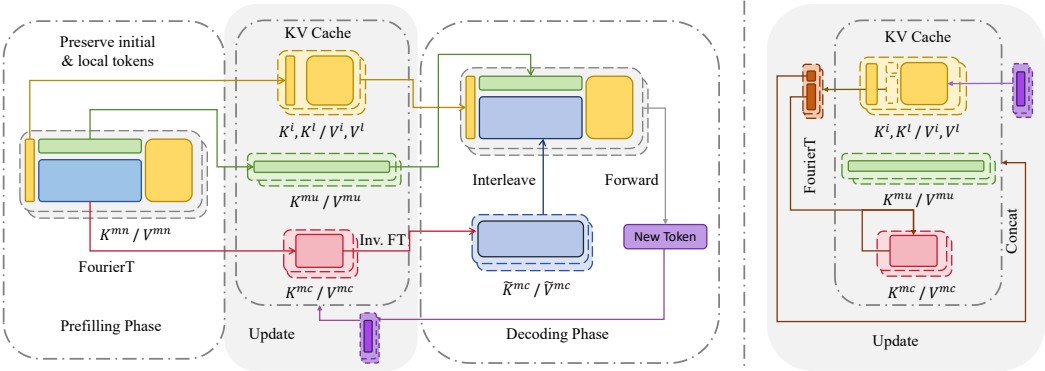

Figure 1: Overview of FourierAttention.

Building on these insights, we introduce **FourierAttention**, a training-free KV cache compression framework using a translated Fourier transform, as shown in Figure 1. Departing from prior methods that uniformly process all head dimensions, FourierAttention identifies localized, context-insensitive dimensions in KV states and approximates their temporal evolution via a fixed set of orthogonal Fourier basis functions. By retaining only the dominant Fourier coefficients ($N \ll L$, where $L$ is the sequence length), our method projects sequences into a compact spectral representation. Unlike polynomial bases that are widely used in HiPPO, FourierAttention exploits the shift-invariance and temporal parallelism of Fourier transforms, allowing for efficient computation and higher performance. During decoding, a customized Triton kernel **FlashFourierAttention** is used to decompose KV cache states during attention calculation, minimizing memory overhead via streamlined read-write operations. Our contributions can be summarized as follows:

- We reveal a bifurcation in Transformer head dimensions: lower dimensions prioritize local context, while upper ones capture long-range dependencies. This inspires us to compress long-context-insensitive dimensions without sacrificing contextual awareness.

- We introduce FourierAttention, which optimizes KV cache by projecting its temporal evolution onto a fixed set of orthogonal Fourier bases. This method efficiently eliminates redundant components while preserving contextual fidelity, achieving a balance between memory and computational efficiency.

- We evaluate FourierAttention's performance on the LLaMA Series using LongBench and NIAH. Our FourierAttention achieves the consistent superiority of long-context performance over other cache optimization methods while maintaining lower memory consumption.

## 2 RELATED WORK

KV cache optimization is a crucial technique for enhancing efficiency in attention-based LLMs (Fu, 2024; Liu et al., 2025). As context length increases, the KV cache in LLMs grows linearly, creating substantial memory overhead that becomes a bottleneck for long-context applications. Beyond architectural modifications during pretraining (Ainslie et al., 2023; Liu et al., 2024a), existing training-free optimization approaches mainly involve token eviction or compression. The former discards tokens based on positional or attention patterns, including StreamingLLM (Xiao et al., 2024), H2O (Zhang et al., 2023), SnapKV (Li et al., 2024b), and PyramidKV (Cai et al., 2024), while the latter compresses KV cache through quantization or low-rank projection, such as KIVI (Liu et al., 2024c), KVQuant (Hooper et al., 2024), and Palu (Chang et al., 2024). However, these methods lack fine-grained consideration of different head dimensions in the KV cache, applying uniform optimization across all dimensions. In contrast, our FourierAttention compresses most dimensions to a fixed length while preserving long-context-sensitive dimensions, effectively reducing KV cache size while maintaining the original long-context capabilities.

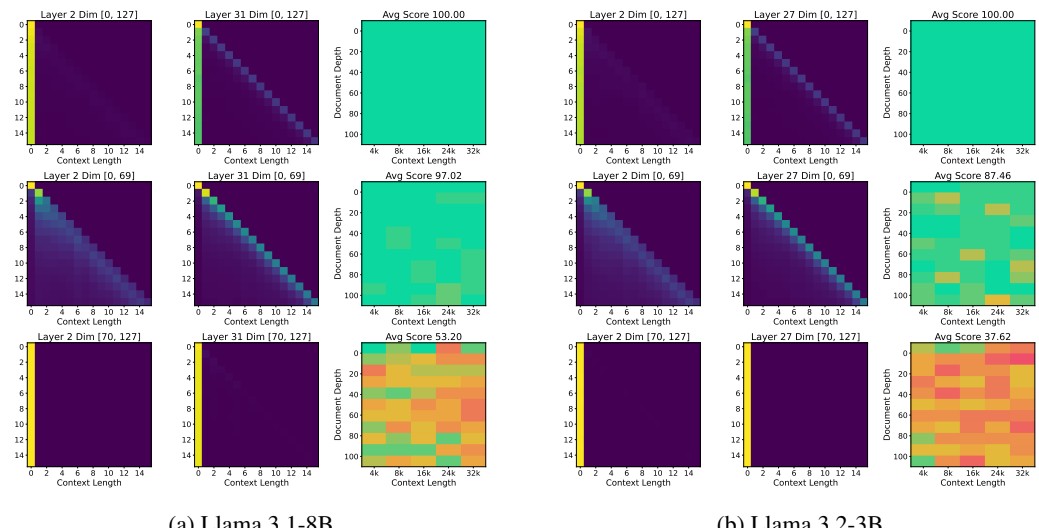

(a) Llama 3.1-8B            (b) Llama 3.2-3B

Figure 2: Visualization of the average attention score and its components in Llama 3.1-8B and Llama 3.2-3B over 32 sentences with 16 tokens. The component of the lower dimensions corresponds to the local branch in Xiao et al. (2024), while that of the upper dimensions corresponds to the global branch. This reveals the different functions of different dimensions in the attention mechanism. It can be further validated that adding Gaussian noise to the lower dimensions has little effect on NIAH performance, but adding noise to the upper dimensions will harm the performance remarkably.

# 3 METHODOLOGY

## 3.1 HEAD DIMENSION HETEROGENEITY

We analyze the heterogeneous sensitivity of transformer head dimensions to varying context lengths. By visualizing attention scores across 128 dimensions in LLaMA architecture (Figure.2), we identify a bifurcation in attention patterns: the first 70 dimensions (0–69) exhibit sharp focus on short-range context, with score distributions concentrated on recent tokens, while the latter 58 dimensions (70–127) maintain a persistent bias toward initial "sink tokens"—positional embeddings that serve as static reference points. This divergence suggests distinct contextual roles encoded within head dimensions, where specialized subsets prioritize local versus global signal retention.

To further validate this hypothesis, we evaluate the model on a Needle-In-A-Haystack retrieval task across sequences of up to 32,000 tokens. As shown in Figure 2a, the baseline model achieves perfect retrieval accuracy (100.0). Introducing Gaussian noise to the first 70 dimensions, performance remains robust (97.02), confirming their limited role in long-range dependency resolution. Conversely, perturbing the latter 58 dimensions catastrophically reduces accuracy to 53.20 on average, with failures consistent across all tested depths and context lengths (Figure 2b mirrors this trend). This stark contrast empirically demonstrates that upper dimensions in transformers are indispensable for retaining long-range information, while lower dimensions specialize in local context encoding. These findings provide critical insights for optimizing memory-efficient architectures, as strategically prioritizing dimensions specialized in long-range retention enhances contextual awareness within memory limits. For more details on dimension selection, please refer to Section 3.4. Due to the heterogeneity further illustrated in Appendix B, we propose **FourierAttention**. In FourierAttention, most dimensions of the KV cache are compressed to a fixed length via translated Fourier transform.

## 3.2 PRELIMINARY: HiPPO FRAMEWORK

Inspired by HiPPO (Gu et al., 2020), we compress these less context-sensitive dimensions into fixed-length states to reduce KV cache storage. Under the HiPPO framework, an infinitely long sequence, $f_{1\cdots L}$, can be approximated by finite-length states, $c \in \mathbb{R}^N$, as the combining coefficients of finite-order basis functions. HiPPO designs different state update equations for various basis

functions under different measure functions, such as LegT based on Legendre Polynomials in a translated fixed window size. Among these methods, FourierT measure based on Translated Fourier Transform is most suitable for token-wise parallelism in transformers, because it can be expressed in matrix form and performed independently in different order states. Therefore, we adopt FourierT to compress cache, $\boldsymbol{K}, \boldsymbol{V} \in \mathbb{R}^{L \times d}$, which also achieves better downstream performance in Section 5.1.

### 3.3 ONLINE COMPRESSION VIA HiPPO-FOURIERT

We set the translated window length in FourierT to the maximum context length, ensuring effective compression within valid input-output ranges. In the prefilling phase, we preserve all dimensions of the initial $L_{\text{init}}$ and the local $L_{\text{local}}$ tokens,

$$
\begin{aligned}
\boldsymbol{K}^i, \boldsymbol{K}^l &= \boldsymbol{K}[: L_{\text{init}}], \boldsymbol{K}[-L_{\text{local}} :] \\
\boldsymbol{V}^i, \boldsymbol{V}^l &= \boldsymbol{V}[: L_{\text{init}}], \boldsymbol{V}[-L_{\text{local}} :]
\end{aligned} \tag{1}
$$

and distinguish the dimension indices $\mathcal{D}^{ku}, \mathcal{D}^{kc}, \mathcal{D}^{vu}, \mathcal{D}^{vc}$ in KV cache for uncompressing and compressing, to enable training-free integration.

$$
\begin{aligned}
\boldsymbol{K}^{mn} &= \boldsymbol{K}[L_{\text{init}} : -L_{\text{local}}, \mathcal{D}^{kc}], \quad \boldsymbol{V}^{mn} = \boldsymbol{V}[L_{\text{init}} : -L_{\text{local}}, \mathcal{D}^{vc}], \\
\boldsymbol{K}^{mu} &= \boldsymbol{K}[L_{\text{init}} : -L_{\text{local}}, \mathcal{D}^{ku}], \quad \boldsymbol{V}^{mu} = \boldsymbol{V}[L_{\text{init}} : -L_{\text{local}}, \mathcal{D}^{vu}].
\end{aligned} \tag{2}
$$

We preserve $\boldsymbol{K}^{mu}, \boldsymbol{V}^{mu}$, compress $\boldsymbol{K}^{mn}, \boldsymbol{V}^{mn}$ to fix-sized $\boldsymbol{K}^{mc} \in \mathbb{R}^{2N \times |\mathcal{D}^{kc}|}, \boldsymbol{V}^{mc} \in \mathbb{R}^{2N \times |\mathcal{D}^{vc}|}$ and use the original KV for forward propagation.

$$
\boldsymbol{K}^{mc} = \frac{1}{T} \boldsymbol{\mathcal{F}} \boldsymbol{K}^{mn}, \quad \boldsymbol{V}^{mc} = \frac{1}{T} \boldsymbol{\mathcal{F}} \boldsymbol{V}^{mn}, \tag{3}
$$

$$
\boldsymbol{O} = \texttt{flash\_attention}(\boldsymbol{Q}, \boldsymbol{K}, \boldsymbol{V}).
$$

We detail the mathematical derivation in Appendix C. When it comes to the compression matrix in FourierT, originally, $\boldsymbol{\mathcal{F}} \in \mathbb{C}^{N \times L_{\text{middle}}}$, where $L_{\text{middle}} = L - L_{\text{init}} - L_{\text{local}}$ and $\boldsymbol{\mathcal{F}}_{nt} = e^{i \frac{2\pi nt}{T}}$. However, since caches in mainstream LLMs are real-valued, we convert complex numbers to corresponding 2D vectors, transforming $N$-order complex states into $2N$-order real states. We denode $N' = 2N$, and the real compression matrix in FourierT is $\boldsymbol{\mathcal{F}} \in \mathbb{R}^{N' \times L_{\text{middle}}}$ as shown in Equation 4.

$$
\boldsymbol{\mathcal{F}} = \begin{bmatrix}
1 & 1 & \cdots & 1 \\
0 & 0 & \cdots & 0 \\
1 & \cos \frac{2\pi}{T} & \cdots & \cos \frac{2\pi(L_{\text{middle}}-1)}{T} \\
0 & \sin \frac{2\pi}{T} & \cdots & \sin \frac{2\pi(L_{\text{middle}}-1)}{T} \\
\vdots & \vdots & \ddots & \vdots \\
1 & \cos \frac{2\pi(N-1)}{T} & \cdots & \cos \frac{2\pi(N-1)(L_{\text{middle}}-1)}{T} \\
0 & \sin \frac{2\pi(N-1)}{T} & \cdots & \sin \frac{2\pi(N-1)(L_{\text{middle}}-1)}{T}
\end{bmatrix}. \tag{4}
$$

In the decoding phase, FourierAttention reconstructs intermediate cache $\tilde{\boldsymbol{K}}^m, \tilde{\boldsymbol{V}}^m$ via inverse Fourier transform in attention computation with the current query vector $\boldsymbol{q}_{t+1}$,

$$
\begin{aligned}
\tilde{\boldsymbol{K}}^m[\mathcal{D}^{ku}] &\leftarrow \boldsymbol{K}^{mu}, \quad \tilde{\boldsymbol{K}}^m[\mathcal{D}^{kc}] \leftarrow \boldsymbol{\mathcal{F}}^T \boldsymbol{K}^{mc}, \quad \tilde{\boldsymbol{K}} = \texttt{cat}(\boldsymbol{K}^i, \tilde{\boldsymbol{K}}^m, \boldsymbol{K}^l) \\
\tilde{\boldsymbol{V}}^m[\mathcal{D}^{vu}] &\leftarrow \boldsymbol{V}^{mu}, \quad \tilde{\boldsymbol{V}}^m[\mathcal{D}^{vc}] \leftarrow \boldsymbol{\mathcal{F}}^T \boldsymbol{K}^{mc}, \quad \tilde{\boldsymbol{V}} = \texttt{cat}(\boldsymbol{V}^i, \tilde{\boldsymbol{V}}^m, \boldsymbol{V}^l)
\end{aligned} \tag{5}
$$

$$
\boldsymbol{o}_{t+1} = \texttt{flash\_attention}(\boldsymbol{q}_{t+1}, \tilde{\boldsymbol{K}}, \tilde{\boldsymbol{V}}).
$$

and compresses tokens out of the local range individually,

$$
\boldsymbol{K}^{mc} \leftarrow \boldsymbol{K}^{mc} + \frac{1}{T} \boldsymbol{f}_{L_{\text{middle}}+1} \boldsymbol{K}^l[0, \mathcal{D}^{kc}], \quad \boldsymbol{V}^{mc} \leftarrow \boldsymbol{V}^{mc} + \frac{1}{T} \boldsymbol{f}_{L_{\text{middle}}+1} \boldsymbol{V}^l[0, \mathcal{D}^{vc}]
$$

$$
\boldsymbol{f}_{t+1} = \begin{bmatrix} 0 & 1 & \cdots & \cos \frac{2\pi(N-1)L_{\text{middle}}}{T} & \sin \frac{2\pi(N-1)L_{\text{middle}}}{T} \end{bmatrix}^\top \tag{6}
$$

To eliminate intermediate read-write cost in decompression, we try to implement a custom kernel, **FlashFourierAttention**, using Triton (Tillet et al., 2019), integrating the decompression into standard FlashAttention2 (Dao, 2024) and FlashDecoding (Dao et al., 2023). FlashFourierAttention loads compressed intermediate states once and decompresses at corresponding sequence positions during iterative KV cache loading, which is further detailed in Appendix D.

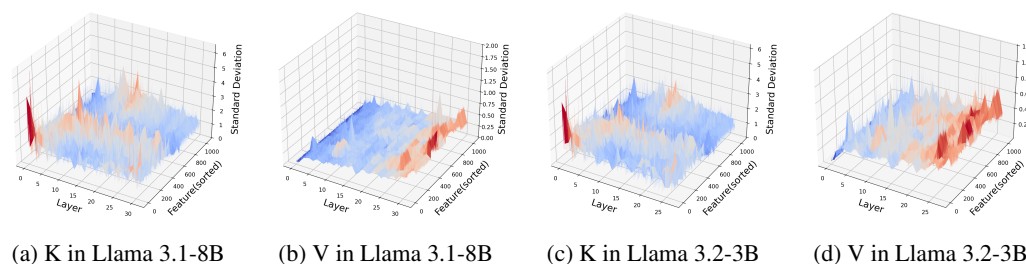

(a) K in Llama 3.1-8B  (b) V in Llama 3.1-8B  (c) K in Llama 3.2-3B  (d) V in Llama 3.2-3B

Figure 3: Visualization of standard deviation of KV cache in different layers in Llama 3.1-8B and Llama 3.2-3B. The feature dimensions are sorted based on the indices in each head.

### 3.4 FINE-GRAINED COMPRESSION SCHEMA

In FourierAttention, a crucial point lies in how to select the dimension to be compressed. To address this, we directly compress and decompress all KV caches, prioritizing dimensions with smaller mean-squared error in reconstruction to a fixed length. Based on further observations of the KV cache, we adopt a fine-grained compression schema, where more dimensions of the V cache and lower-layer caches are compressed to a fixed length. We analyze the standard deviation of KV cache dimensions along the temporal direction across different layers and find that for both Llama 3.1-8B and Llama 3.2-3B, as shown in Figure 7. The standard deviation of the K cache is consistently higher than that of the V cache, and the standard deviation in upper layers exceeds that of lower layers. Consequently, we compress more dimensions of the smoother V cache and lower-layer caches to a fixed length, while retaining more K cache and upper-layer caches to extend with sequence length. Thus, FourierAttention exhibits an asymmetric, inverted-pyramid compression pattern.

Interestingly, this differs from most KV cache compression approaches. Works like Cai et al. (2024) and Xing et al. (2024) suggest preserving more KV caches in lower layers, as attention becomes sparser in upper layers. However, in FourierAttention, the optimization criterion is whether the dimension can be well reconstructed. Since caches in upper layers exhibit more oscillatory features due to more deterministic predictions, we retain more dimensions to maintain output stability.

## 4 EXPERIMENT

### 4.1 SETUP

We conduct experiments on Llama 3.1-8B (Dubey et al., 2024) and Llama 3.2-3B (Meta, 2024a). For all models, we set the length of initial tokens $L_{\text{init}}$ to 4, the length of local tokens $L_{\text{local}}$ to 1024, and the number of states $N = 512$, namely $N' = 1024$. We evaluate the reconstruction loss using the prompt portion of the 32k Needle-In-A-Haystack benchmark in OpenCompass (Contributors, 2023). As mentioned earlier, we employ an asymmetric inverted pyramid compression strategy: for the first 4 layers, we compress 90% of K dimensions and 95% of V dimensions; for the last 8 layers, 50% of K and 70% of V; and for the remaining layers, 80% of both K and V. Overall, 76% KV caches are compressed to a fixed length. All experiments are performed on an NVIDIA H100 GPU with FP16 precision and accelerated with FlashAttention2 (Dao, 2024).

### 4.2 LONG-CONTEXT EVALUATION

We evaluate our method against other KV cache optimization approaches with two long-context benchmarks in OpenCompass (Contributors, 2023), LongBench (Bai et al., 2023), and Needle-In-A-Haystack (NIAH) (Kamradt, 2023; Li et al., 2024a), with a truncation context length of 32K. We compare with StreamingLLM (Xiao et al., 2024), SnapKV (Li et al., 2024b), PyramidKV (Cai et al., 2024), and Palu (Chang et al., 2024), covering both token eviction and feature compression. For fair comparison, we retain 4 initial tokens and 1024 local tokens in StreamingLLM, additionally keep 1024 recalled middle tokens, matching our compressed dimension count, in SnapKV and PyramidKV, and compress KV feature dimensions to 70% in Palu.

| | **Single-Doc** | | | **Multi-Doc** | | | **Summary** | | | **Few-shot** | | | **Synthetic** | | **Code** | | **Avg.** |
|---|---|---|---|---|---|---|---|---|---|---|---|---|---|---|---|---|---|
| | NQ | Qsp | MF | HQ | WQ | Msq | GR | QS | MN | TR | TQ | SS | PC | PR | LCC | Re-P | |
| **Llama-3.1-8B** | 13.2 | 20.2 | 32.8 | 12.0 | 13.6 | 8.7 | 29.7 | 25.1 | 0.9 | 73.5 | 91.0 | 47.3 | 0.8 | 26.8 | 72.0 | 69.3 | 39.5 |
| + SLM | 7.9 | 13.9 | 15.6 | 7.8 | 10.1 | 4.5 | 19.9 | 21.5 | 9.9 | 61.5 | 84.7 | 43.5 | 1.3 | 6.2 | 58.4 | 56.4 | 31.5 |
| + SnapKV | 12.7 | 19.8 | 32.5 | 12.0 | 13.8 | 8.6 | 29.2 | 24.9 | 12.6 | 73.0 | 91.0 | 46.5 | 0.8 | 26.8 | 60.0 | 59.7 | 37.1 |
| + PyramidKV | 18.5 | 19.8 | 32.5 | 12.1 | 13.8 | 8.7 | 29.7 | 24.9 | 12.4 | 73.0 | 90.1 | 46.6 | 0.8 | 26.8 | 60.0 | 59.4 | 37.3 |
| + Palu | 4.5 | 18.0 | 21.6 | 9.5 | 11.3 | 5.2 | 17.3 | 6.9 | 9.0 | 68.5 | 83.4 | 32.3 | 0.6 | 14.6 | 56.7 | 54.6 | 30.7 |
| + FA (ours) | 15.6 | 19.8 | 32.7 | 12.0 | 13.5 | 7.9 | 24.1 | 24.0 | 0.7 | 73.0 | 91.2 | 46.0 | 1.1 | 26.9 | 71.4 | 66.3 | **38.6** |
| **Llama-3.2-3B** | 10.3 | 21.7 | 35.5 | 9.6 | 12.8 | 6.8 | 30.2 | 23.8 | 28.2 | 70.0 | 87.2 | 38.2 | 0.0 | 7.0 | 70.0 | 66.4 | 38.0 |
| + SLM | 9.1 | 17.6 | 21.6 | 7.1 | 9.8 | 4.0 | 19.0 | 21.3 | 23.2 | 53.0 | 84.3 | 39.8 | 1.4 | 6.5 | 55.5 | 53.5 | 31.2 |
| + SnapKV | 9.4 | 21.0 | 35.0 | 9.5 | 12.8 | 6.6 | 29.5 | 23.4 | 27.8 | 69.5 | 86.4 | 38.3 | 0.0 | 6.8 | 58.2 | 56.4 | 34.9 |
| + PyramidKV | 8.9 | 21.4 | 35.7 | 9.5 | 12.8 | 6.8 | 30.4 | 23.6 | 28.2 | 69.5 | 86.9 | 38.7 | 0.0 | 6.8 | 58.7 | 57.3 | 35.2 |
| + Palu | 2.0 | 19.2 | 20.4 | 5.8 | 10.3 | 2.7 | 13.4 | 4.1 | 14.0 | 57.0 | 47.4 | 21.5 | 1.5 | 3.1 | 55.1 | 49.5 | 25.5 |
| + FA (ours) | 12.8 | 21.1 | 35.8 | 9.8 | 11.6 | 6.5 | 23.0 | 23.8 | 24.1 | 69.0 | 87.0 | 39.4 | 0.0 | 6.2 | 69.8 | 63.3 | **37.0** |

Table 1: Results of LLaMA Series (Dubey et al., 2024; Meta, 2024b) on LongBench (Bai et al., 2023). Our FourierAttention (FA) achieves consistent superiority over StreamingLLM (SLM), SnapKV, PyramidKV, and Palu and shows the closest performance with LLMs with full attention.

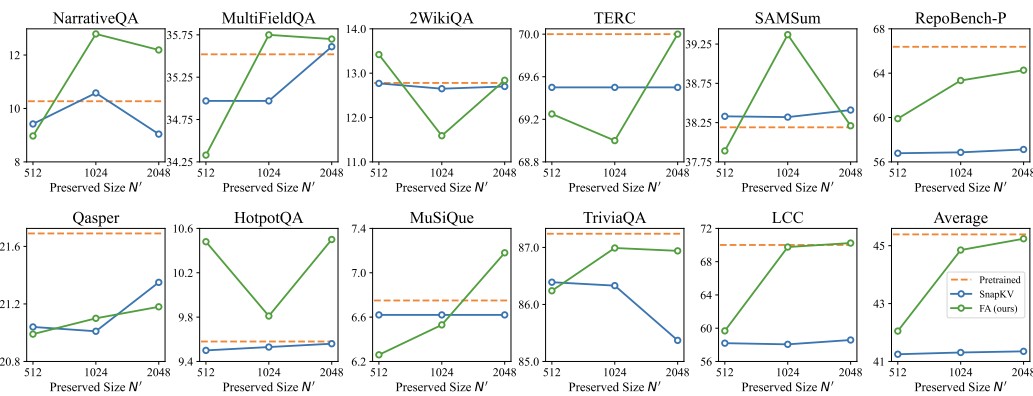

Figure 4: Comparison of different preserved sizes $N'$ between FourierAttention and SnapKV in Llama 3.2-3B (Meta, 2024a). FourierAttention outperforms SnapKV on average across different $N'$.

For LongBench as shown in Tables 1, our FourierAttention achieves performance closest to the original model and shows consistent superiority over other cache optimization methods on both Llama 3.2-3B and Llama 3.1-8B. For the NIAH task as shown in Figure 5 and 6, we similarly achieve performance closest to the original pretrained models at 32k context length. While SnapKV and PyramidKV are theoretically suitable for retrieval tasks like NIAH, they still exhibit recall errors. Though Palu maintains stable attention approximation under moderate compression, 30-50%, they show significant performance degradation at 75% compression due to insufficient granular analysis of KV cache features. In contrast, our FourierAttention optimizes compression by identifying and preserving KV dimensions insensitive to compression, thereby maximally retaining the long-context capabilities and demonstrating superiority across both models and benchmarks.

On Llama 3.2-3B, we also verify the downstream task performance of FourierAttention with different values of $N'$ and SnapKV with the corresponding intermediate recall sizes, as shown in Figure 4. We find that, regardless of the value of $N'$, FourierAttention exhibits sufficient robustness and outperforms SnapKV on average across different recall sizes. We chose $N' = 1024$ in our paper, which is a parameter that balances performance and speed.

## 4.3 EFFICIENCY VALIDATION

In addition to comparisons in downstream performance, we also conduct an efficiency comparison of FourierAttention with SnapKV and Palu, two representative cache optimization methods. First, in

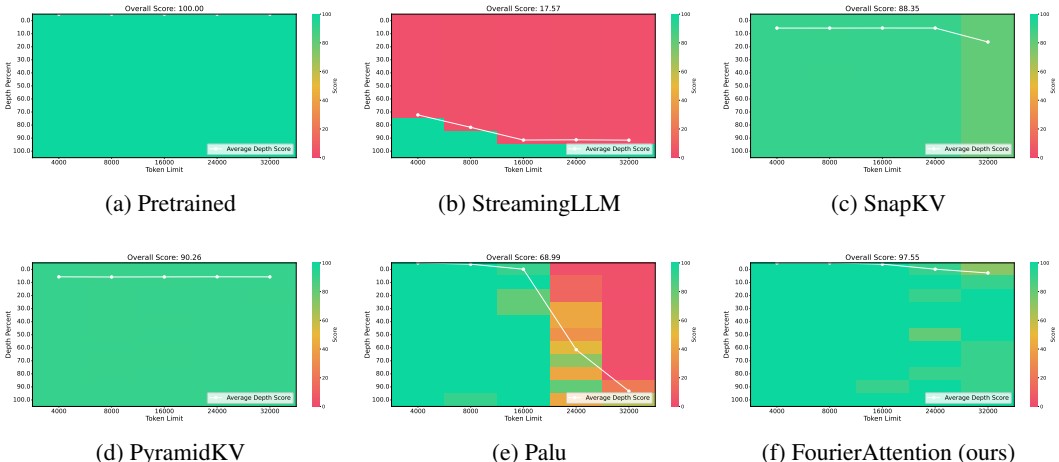

Figure 5: Results of Llama 3.1-8B (Dubey et al., 2024) on Needle-In-A-Haystack (Kamradt, 2023). FourierAttention achieves the highest average score over StreamingLLM, SnapKV, PyramidKV, and Palu, and shows the closest performance with LLMs with full attention.

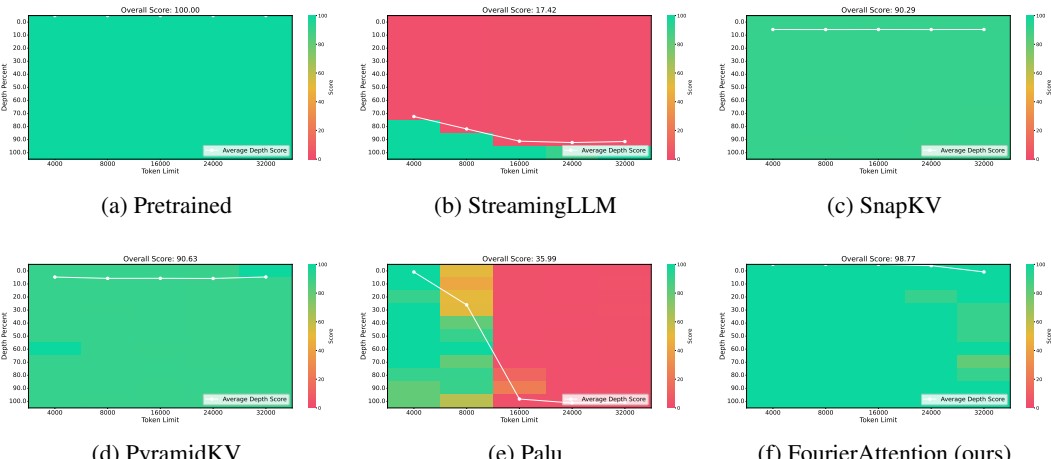

Figure 6: Results of Llama 3.2-3B (Meta, 2024a) on Needle-In-A-Haystack (Kamradt, 2023). FourierAttention achieves achieves the highest average score over StreamingLLM, SnapKV, PyramidKV, and Palu, and shows the closest performance with LLMs with full attention.

terms of storage, thanks to the custom Triton kernel, FlashFourierAttention, detailed in Appendix D, compared with SnapKV and Palu, we achieve a clear advantage in memory efficiency and 80k context length inference of Llama 3.1-8B and Llama 3.2-3B on a single H100. In the prefilling phase, since we only add one Fourier transform, matrix multiplication, compared with the original attention and use the efficient FFT built-in operator, we have achieved a latency close to that of existing efficient approaches such as Palu and SnapKV in long contexts ranging from 16k to 80k. In addition, thanks to the accelerated operators during the decoding phase, we achieve throughput comparable to SnapKV.

## 5 DISCUSSION

### 5.1 CHOICE OF BASIS FUNCTIONS

Although FourierAttention employs HiPPO-FourierT for compression, Gu et al. (2020) proposes and claims polynomial basis functions like LegT with superior performance. While maintaining identical sliding window sizes, we compare LegT and FourierT in reconstructing KV caches from LLMs. As illustrated in Figure 8, we evaluate their reconstruction effects on 4 randomly selected KV cache

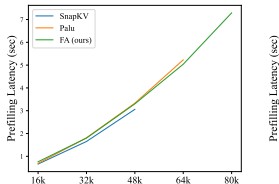 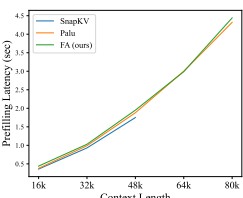 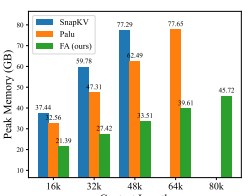 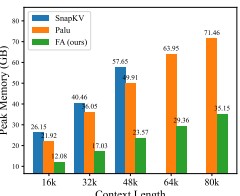

(a) Llama 3.1-8B Latency  (b) Llama 3.2-3B Latency  (c) Llama 3.1-8B Memory  (d) Llama 3.2-3B Memory

Figure 7: Visualization of standard deviation of KV cache in different layers in Llama 3.1-8B and Llama 3.2-3B. The feature dimensions are sorted based on the indices in each head.

| | **NIAH** | | | | | | | | **AGG** | | | **QA** | | **Avg.** |
|---|---|---|---|---|---|---|---|---|---|---|---|---|---|---|
| | SK1 | SK2 | SK3 | MK1 | MK2 | MK3 | MV | MQ | CWE | FWE | VT | SQ | HP | |
| **Llama 3.2-3B** | 100.0 | 100.0 | 100.0 | 99.0 | 100.0 | 99.0 | 100.0 | 99.8 | 60.0 | 89.7 | 97.0 | 77.0 | 53.0 | 90.3 |
| + FourierT | 100.0 | 100.0 | 98.0 | 99.0 | 99.0 | 100.0 | 96.0 | 97.8 | 57.9 | 80.7 | 76.6 | 81.0 | 51.0 | **87.5** |
| + LegT | 55.0 | 82.0 | 42.0 | 89.0 | 93.0 | 50.0 | 93.8 | 84.0 | 13.5 | 58.3 | 76.0 | 36.0 | 21.0 | 61.0 |
| + uniform | 100.0 | 100.0 | 99.0 | 99.0 | 99.0 | 98.0 | 93.0 | 98.8 | 59.7 | 77.3 | 78.2 | 80.0 | 50.0 | 87.1 |
| + KV inv. | 100.0 | 100.0 | 100.0 | 99.0 | 99.0 | 100.0 | 93.5 | 98.8 | 54.3 | 79.7 | 72.6 | 80.0 | 53.0 | 86.9 |
| + layer inv. | 100.0 | 98.0 | 88.0 | 98.0 | 97.0 | 94.0 | 80.0 | 96.8 | 41.3 | 69.3 | 61.4 | 80.0 | 50.0 | 81.1 |

Table 2: Validation of basis function and compression schema in Llama 3.2-3B based on RULER in 4k context length. Besides AGG tasks, FourierAttention achieve performance close to original model.

dimensions from layer 0 of Llama 3.2. Under equivalent state dimensions[1], FourierT consistently achieves lower reconstruction loss than LegT.

We further evaluate FourierT and LegT compression on Llama 3.2-3B using more discriminative RULER benchmark (Hsieh et al., 2024) in 4k context length. For fair comparison, we employ the same method to identify dimensions suitable for LegT compression and apply an identical compression schema. Results in Table 2 show FourierT still performs better, demonstrating that FourierT offers better parallelizability for compression efficiency and performance in downstream evaluation. Nevertheless, we must acknowledge that FourierAttention performs slightly behind the pre-trained model on aggregation (AGG) tasks, such as VT. We attribute this to the fact that aggregation tasks rely on statistical information and demand finer input details. By discarding high frequencies during compression, FourierAttention loses these details and thus degrades on such tasks.

## 5.2 ABLATION ON COMPRESSION SCHEMA

As mentioned in Section 3.4, we propose a more fine-grained compression scheme based on additional observations of the KV cache. As shown in the Table 2, we compare three approaches: uniform compression across all layers and between KV (uniform), inverted KV compression schema by K-priority over V (KV inv.), and inverted layer-wise compression schema by upper-layer priority over lower-layer (layer inv.). Results demonstrate that our original V-priority and lower-layer-priority compression schema achieves superior performance on RULER in a 4k context length. This further illustrates that frequency-based sequence-wise KV cache compression exhibits different optimization characteristics compared to conventional KV token eviction (Cai et al., 2024; Xing et al., 2024).

## 5.3 COMPRESSED DIMENSION DISTRIBUTION

Finally, we analyze the compressed dimensions selected by our FourierAttention. We count the number of each dimension selected for compression, averaged across attention heads in different layers, grouped every 16 dimensions. Results in Figure 9 show that in both Llama 3.1-8B and Llama 3.2-3B, starting from layer 2, lower dimensions are more frequently compressed while the upper dimensions tend to preserve complete temporal information in our FourierAttention. This

---

[1]FourierT uses lower-order basis functions since FourierAttention's state size is twice the number of states

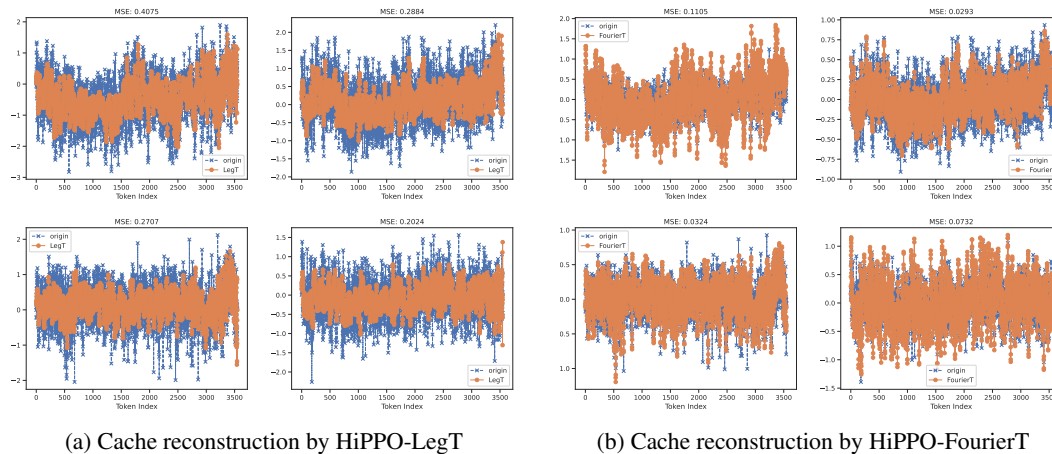

(a) Cache reconstruction by HiPPO-LegT   (b) Cache reconstruction by HiPPO-FourierT

Figure 8: Visualization of KV cache reconstruction in Llama 3.2-3B for different basis functions, LegT and FourierT under HiPPO framework (Gu et al., 2020). FourierT outperforms LegT in cache reconstruction with lower reconstruction loss.

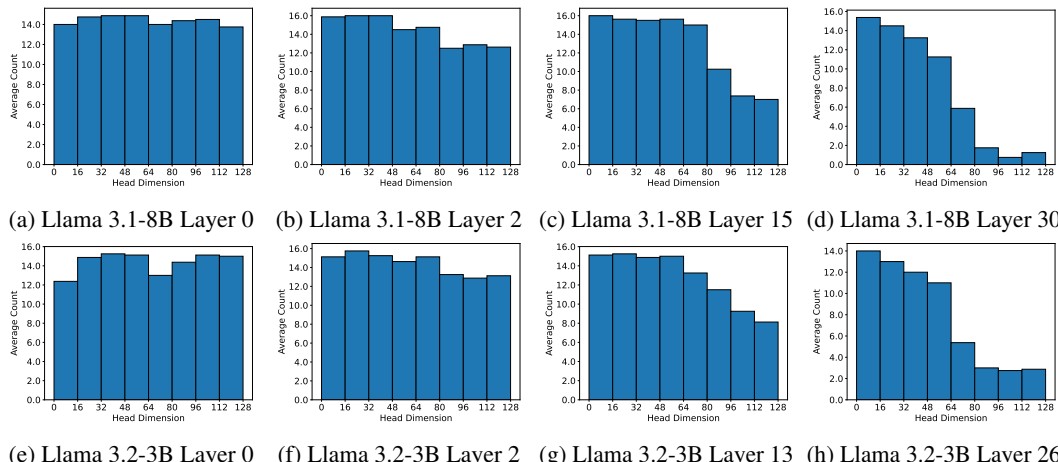

(a) Llama 3.1-8B Layer 0 (b) Llama 3.1-8B Layer 2 (c) Llama 3.1-8B Layer 15 (d) Llama 3.1-8B Layer 30

(e) Llama 3.2-3B Layer 0 (f) Llama 3.2-3B Layer 2 (g) Llama 3.2-3B Layer 13 (h) Llama 3.2-3B Layer 26

Figure 9: The statistics of each dimension selected for compression, averaged across attention heads in different layers, grouped every 16 dimensions, in Llama 3.1-8B and Llama 3.2-3B.

phenomenon is more evident in upper layers, where fewer dimensions are chosen to be compressed. As illustrated in Figure 2, these uncompressed upper dimensions primarily contribute to forming attention sinks and capturing long-context semantic relationships, thus requiring complete retention, whereas other dimensions can be stored with limited length.

# 6 CONCLUSION

We propose FourierAttention, a novel KV cache optimization approach that compresses long-context-insensitive dimensions without sacrificing contextual awareness based on an interesting phenomenon in transformer head dimensions, that lower dimensions capture local features, while upper ones capture long-context dependencies. Inspired by HiPPO, we optimize the long-context-insensitive KV cache through a translated Fourier transform into fixed-length states in the prefilling phase and reconstruct the KV cache in the decoding phase. FourierAttention shows the best performance on the LLaMA Series in LongBench and NIAH on average. We are trying to improve the efficiency of FourierAttention through a customized Triton-based kernel, FlashFourierAttention, eliminating intermediate read-write operations and effectively reducing memory overhead.

ETHICAL STATEMENT

This research adheres to established ethical standards. To the best of our knowledge, our study does not process any sensitive personal data, does not involve any human subjects, and does not target any ethically risky applications. All experiments and analyses are conducted in line with recognized guidelines, ensuring integrity, transparency, and reliability.

REPRODUCIBILITY STATEMENT

To ensure the reproducibility of and to support the open-source community, we will publicly release FourierAttention, its trained checkpoints, and the complete training and evaluation code, especially the custom Triton kernel. We expect these as a reference for future work on long-context LLMs, facilitating innovation and advancing progress in this field.

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

## A    USE OF LARGE LANGUAGE MODELS

We only use Large Language Models for language-centric assistance, namely, for grammar, style, and clarity, ensuring that no component of research ideation, experimental design, or scientific contribution is either influenced or generated by LLM outputs.

## B    MORE OBSERVATION ON HETEROGENEITY

Regarding the observation on K cache, there exists a relation between this dimension bifurcation with RoPE. This is a very correct insight. In Figure 2, we compare the attention score components of the first 70 dimensions and the last 58 dimensions of all QK states in Llama 3.1-8B and Llama 3.2-3B, as well as the effects on the downstream NIAH task after adding noise. The choice of the first 70 dimensions refers to the concept of critical dimension in Liu et al. (2024b), which corresponds to the number of dimensions with complete position information observed during the pre-training phase in LLMs with RoPE. This dimension $d_{\text{extra}}$ can be calculated based on the head dimension $d$, the initial pre-training length $T_{\text{train}}$, and the initial rotary base $\beta$.

$$d_{\text{extra}} = 2 \left\lceil \frac{d}{2} \log_\beta \frac{T_{\text{train}}}{2\pi} \right\rceil$$

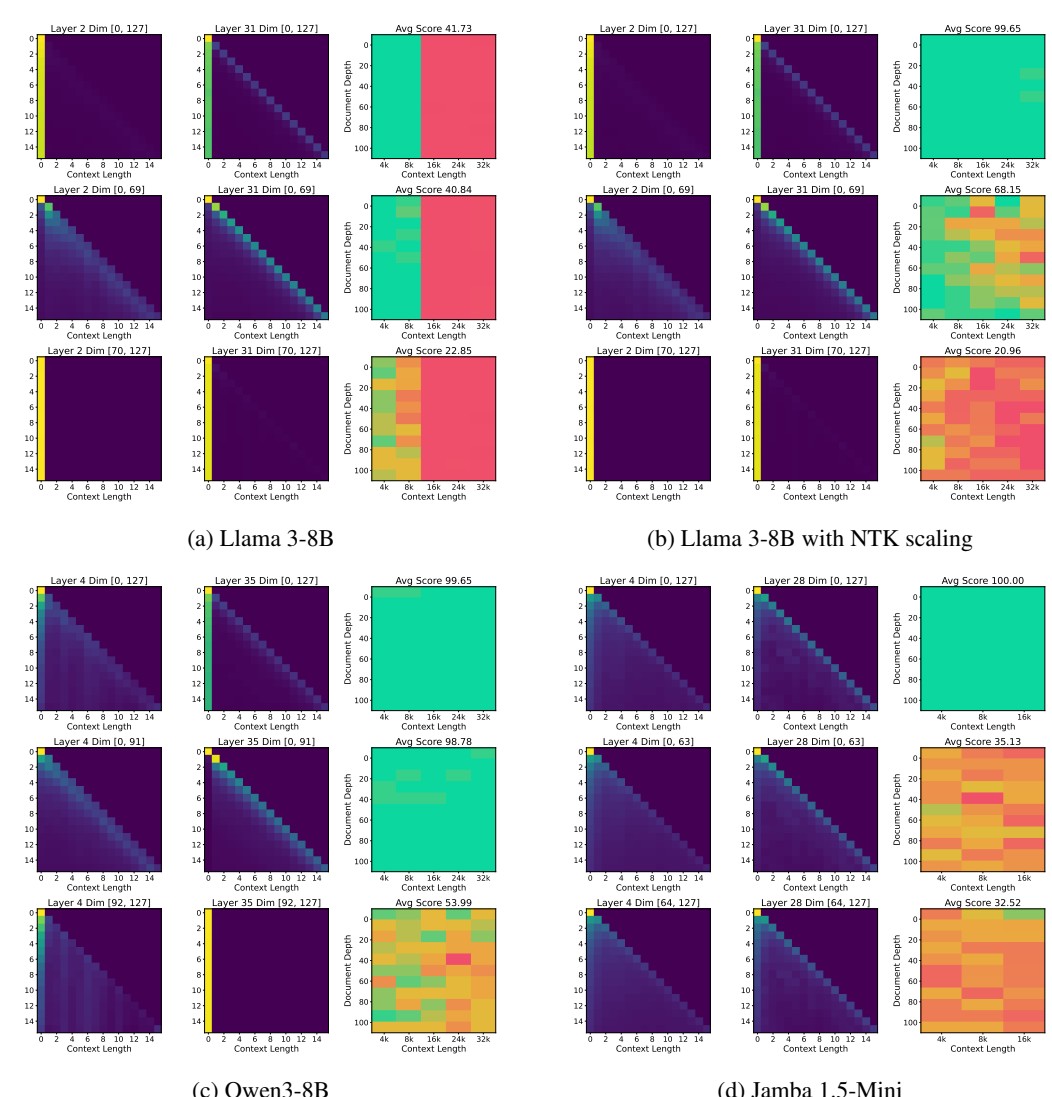

(a) Llama 3-8B

(b) Llama 3-8B with NTK scaling

(c) Qwen3-8B

(d) Jamba 1.5-Mini

Figure 10: Visualization of the average attention score and its components in other RoPE-based LLMs, including Llama 3-8B (Meta, 2024b), as well as its NTK-scaled model, and Qwen3-8B, (Meta, 2024a), and NoPE-based LLMs, such as Jamba 1.5-Mini (Team et al., 2024). In RoPE-based LLMs, the component of the lower dimensions corresponds to the local branch in StreamingLLM (Xiao et al., 2024), while that of the upper dimensions corresponds to the global branch. This reveals the different functions of different dimensions in the attention mechanism. It can be further validated that adding Gaussian noise to the lower dimensions has little effect on NIAH performance, but adding noise to the upper dimensions will harm the performance remarkably. In NoPE-based LLMs, the components of the lower and upper dimensions do not have heterogeneous features in RoPE-based LLMs. Besides, adding Gaussian noise to the lower or upper dimensions will harm the NIAH performance equally.

Some studies have found that the self-attention components before and after the critical dimension have different characteristics in long contexts (Liu et al., 2024b; Wei et al., 2025), and when the rotation angle is large, LLMs are not good at characterizing long-context features (Men et al., 2024). Therefore, we choose the critical dimension as the split and find that different dimensions before and after have different impacts on long-context tasks. For LLaMA3 as well as Llama 3.1 and Llama 3.2, $d = 128$, $T_{\text{train}} = 8192$, $\beta = 500000$, and the calculated critical dimension size is 70.

We have also observed similar conclusions on Llama 3-8B with short contexts as shown in Figure 10a. For the first 70 dimensions and the last 58 dimensions of Llama 3-8B, adding noise to the first 70

dimensions has almost no impact, while adding noise to the last 58 dimensions leads to a significant drop in performance within the pre-training length (8k). As for the case of a larger rotation angle base, we scale up the rotary base of Llama 3-8B by 13× to support a 32k context. Similarly, as shown in Figure 10b, for the NTK-extrapolated Llama 3-8B, adding noise to the first 70 dimensions and the last 58 dimensions, respectively, we find that noise in the first 70 dimensions has a weaker impact, while noise in the last 58 dimensions has a greater impact.

We have also verified our observations on the Qwen3-3B (Yang et al., 2025), as shown in Figure 10c. Since the initial pre-training length and rotation angle base of Qwen3 are 4096 and 10000, respectively, the critical dimension $d_{\text{extra}} = 92$. Then we add noise to the first 92 dimensions and the last 36 dimensions, respectively, and find that noise in the upper dimensions has a greater impact on NIAH.

To prove that RoPE indeed causes the above phenomenon, we also conduct experiments on the hybrid model Jamba (Team et al., 2024) without position encoding, as shown in Figure 10d. Since only 1/8 of the layers in Jamba use self-attention, we increase the noise level to $\mathcal{N}(0, 16)$. We find that for the first 64 dimensions and the last 64 dimensions, adding noise respectively produces effects that are quite close, and not as significant as in LLMs with RoPE. At the same time, we have also observed that in the above RoPE-based LLM, the upper dimensions contribute to attention sink, while the lower dimensions contribute to local attention. These phenomena have not been presented on Jamba.

## C  MATHEMATICAL DERIVATION

We will present the derivation process of HiPPO-FourierT in this section. Firstly, according to the HiPPO framework (Gu et al., 2020), we give the definitions of the basis functions and the measure functions. We use the Fourier bases as basis functions $g_n(x,t)$, and adopt the translated average measure $\mu(x,t)$ within a fixed window. The formulas are as follows, where the length of the fixed window is equal to the maximum context length T supported by the model.

$$g_n(x,t) = e^{i\frac{2\pi n x}{T}}, \quad \mu(x,t) = \frac{1}{T}\mathbb{I}_{[t-T,t]} \tag{7}$$

Based on this, the input signal $f(t)$, which is the KV cache to be compressed, can be obtained as the projection $c_n(t)$ of the basis functions $g_n(x,t)$ under the measure $\mu(x,t)$.

$$\begin{aligned}
c_n(t) &= \langle f_{\leq t}(x), g_n(x,t)\rangle_{\mu(x,t)} \\
&= \int_o^t f(x) \cdot g_n(x,t) \cdot \mu(x,t)\mathrm{d}x
\end{aligned} \tag{8}$$

Then, we differentiate the state $c_n(t)$.

$$\begin{aligned}
\frac{\mathrm{d}c_n(t)}{\mathrm{d}t} &= \int_o^t f(x) \cdot \frac{\partial g_n(x,t)}{\partial t} \cdot \mu(x,t)\mathrm{d}x + \\
&\quad \int_o^t f(x) \cdot g_n(x,t) \cdot \frac{\partial \mu(x,t)}{\partial t}\mathrm{d}x
\end{aligned} \tag{9}$$

Since $\frac{\partial g_n(x,t)}{\partial t} = 0$ and $\frac{\partial \mu(x,t)}{\partial t} = \frac{\delta_t}{T} - \frac{\delta_{t-T}}{T}$, the differentiation of $c_n(t)$ is simplified as follows.

$$\begin{aligned}
\frac{\mathrm{d}c_n(t)}{\mathrm{d}t} &= 0 + \int_o^t f(x) \cdot g_n(x,t) \cdot \frac{\delta_t}{T}\mathrm{d}x - 0 \\
&= \frac{f(t)}{T} \cdot e^{i\frac{2\pi n t}{T}}
\end{aligned} \tag{10}$$

Considering that the actual storage of LLM is real numbers, we calculate the derivatives of the real and imaginary parts, respectively.

$$\begin{aligned}
\mathrm{Re}\left[\frac{\mathrm{d}c_n(t)}{\mathrm{d}t}\right] &= \frac{f(t)}{T}\cos\frac{2\pi n t}{T}, \\
\mathrm{Im}\left[\frac{\mathrm{d}c_n(t)}{\mathrm{d}t}\right] &= \frac{f(t)}{T}\sin\frac{2\pi n t}{T}
\end{aligned} \tag{11}$$

After discretizing them respectively, we obtain the final state update equations. Regarding the choice of discretization strategy, since the $\boldsymbol{A}$ matrix in HiPPO-FourierT is an identity matrix, the results obtained by different discretization methods in HiPPO-FourierT are only different in step size. We choose the simplest forward Euler discretization as follows.

$$c_{t+1}^{(n)} = \begin{cases} c_t^{(n)} + \frac{f_t}{T}\cos\frac{2\pi n t}{T} & n = 2m \\ c_t^{(n)} + \frac{f_t}{T}\sin\frac{2\pi n t}{T} & n = 2m+1 \end{cases} \tag{12}$$
$$m = 0, 1, \cdots, N-1$$

Then we can derive the compression, update, and decompression functions as shown in Equation 4 in our paper,

$$\boldsymbol{c}_{t+1} = \frac{1}{T}\boldsymbol{\mathcal{F}}_{N \times t}\boldsymbol{f}_t$$

$$\boldsymbol{\mathcal{F}}_{k \times t} = \begin{bmatrix} 1 & 1 & \cdots & 1 \\ 0 & 0 & \cdots & 0 \\ 1 & \cos\frac{2\pi}{T} & \cdots & \cos\frac{2\pi t}{T} \\ 0 & \sin\frac{2\pi}{T} & \cdots & \sin\frac{2\pi t}{T} \\ \vdots & \vdots & \ddots & \vdots \\ 1 & \cos\frac{2\pi(N-1)}{T} & \cdots & \cos\frac{2\pi(N-1)t}{T} \\ 0 & \sin\frac{2\pi(N-1)}{T} & \cdots & \sin\frac{2\pi(N-1)t}{T} \end{bmatrix}, \tag{13}$$

where

$$\boldsymbol{c}_{t+1} = \left[c_{t+1}^{(0)}, c_{t+1}^{(1)}, \cdots, c_{t+1}^{(2k-2)}, c_{t+1}^{(2k-1)}\right]^\top.$$
$$\boldsymbol{f}_t = [f_0, f_1, \cdots, f_t]^\top$$

Regarding discretization, since the A matrix in HiPPO-FourierT is a zero matrix, the results obtained by different discretization methods in HiPPO-FourierT are only different in step size. We finally chose the simplest forward Euler discretization. In the comparative experiments, we have compared HiPPO-LegT with the bilinear discretization method, which is the best-performing discretization method reported in Gu et al. (2020).

## D  TRITON IMPLEMENTAION DETAILS

Regarding Triton implementation details, since PyTorch itself already provides efficient support for Fourier transforms, we only need to design and integrate a custom kernel for the decoding stage, focusing on solving the decompression calculations involved in the decoding stage, and wrapping the decompression within the attention calculations to avoid the read-write of full-size KV cache. In this process, we applied three techniques: FlashDecoding, dimension reordering, and compression delay.

FlashDecoding is a fundamental method for decoding acceleration (Dao et al., 2023). Building on the iterative calculation of self-attention in FlashAttention, it targets the parallel computation of attention in the decoding phase. During decoding, it first segments and computes in parallel, and within each segment, it calculates the attention numerators, denominators, and max attention scores in a block-cycling manner. Finally, it merges the results from different segments. Since the inverse Fourier transform is essentially a matrix multiplication that can be split and computed in parallel along the sequential dimension, the decompression logic of FourierAttention can naturally be adapted to FlashDecoding. Each segment loads the complete compressed state, first decompresses the K cache within the corresponding block, concatenates it with the retained part, multiplies QK, and then gets the attention distribution. After that, it decompresses the corresponding V cache, concatenates it with the retained part, and outputs the attention result within the block.

Secondly, the dimension indices of the KV cache that need to be compressed and those that need to be fully retained, denoted as $\mathcal{D}^{ku}, \mathcal{D}^{kc}, \mathcal{D}^{vu}, \mathcal{D}^{vc}$, are theoretically discrete. However, to improve the loading speed, following the approach in Duanmu et al. (2024), we reorder the dimensions of the QK and VO matrices according to these indices. Specifically, the dimensions in $\mathcal{D}^{ku}, \mathcal{D}^{vu}$ that do not need to be compressed are placed at the lower part of the QK features and VO features of each head, respectively. Conversely, the dimensions in $\mathcal{D}^{kc}, \mathcal{D}^{vc}$ that need to be compressed are placed at the

upper part of the QK features and VO features of each head. In addition, we also record the average value and standard deviation of each dimension in the original and reconstructed cache, and use these to standardize, thus maintaining the numeric stability.

Finally, during the decoding process, theoretically, we need to continuously compress the KV cache that exceeds the local length into an intermediate state. However, if the compression process is called every time during decoding, it would lead to a significant waste of computation. Therefore, we set a delayed compression decoding step, and only compress the exceeded local part of the KV cache once after decoding for several steps, reducing the impact of compression on latency.

