# OpenReview forum: "Beyond Homogeneous Attention: Memory-Efficient LLMs via Fourier-Approximated KV Cache"
_ICLR.cc/2026/Conference — Submitted to ICLR 2026_

### Official Review · Reviewer_wpka · 2025-10-22

**Soundness:** 2
**Presentation:** 3
**Contribution:** 2
**Rating:** 4
**Confidence:** 4

**Summary:**

This work is based on an interesting observation that, within attention heads, some dimensions focus on local context while others capture long-range dependencies. To exploit this heterogeneity, the paper introduces FourierAttention, which compresses the locally focused dimensions of the KV cache into fixed-length states using the HiPPO framework combined with a translated Fourier transform, achieving efficient and training-free KV-cache compression.

**Strengths:**

* **Novel Observation.** This work presents an interesting finding that within a single attention head, some dimensions focus on local context while others capture global dependencies. This insight provides a new perspective on KV-cache compression, distinct from prior token-wise, head-wise, or precision-wise (quantization-based) methods.

* **Solid System Implementation.** The paper adapts kernels from FlashAttention and FlashDecoding to build FlashFourierAttention, effectively integrating Fourier-based compression into the attention computation and avoiding extra memory movement during inference.

**Weaknesses:**

* **Limited Latency Evaluation.** The experiments only report prefill latency, while the majority of KV-cache read and update operations occur during decoding. Including TPOT (Time per Output Token) or throughput comparisons would provide a more complete performance evaluation.
* **Limited Model Generalization.** FourierAttention is built upon the finding of heterogeneous dimension roles within attention heads, yet the evaluation is limited to LLaMA-3.1-8B and LLaMA-3.2-3B. Models with different head sizes (e.g., LLaMA-3.2-1B with only 64 dimensions per head) may not exhibit the same behavior, raising questions about general applicability.
* **Lack of Per-Head Analysis.**
  Prior works such as MInference [1], RazorAttention [2], DuoAttention [3], and HeadKV [4] have shown that sparsity and attention patterns vary across heads. This paper provides limited analysis on whether the observed dimension heterogeneity holds consistently across all attention heads, leaving this aspect unclear.



[1] Jiang, Huiqiang, et al. "Minference 1.0: Accelerating pre-filling for long-context llms via dynamic sparse attention." *Advances in Neural Information Processing Systems* 37 (2024): 52481-52515.

[2] Tang, Hanlin, et al. "Razorattention: Efficient kv cache compression through retrieval heads." *arXiv preprint arXiv:2407.15891* (2024).

[3] Xiao, Guangxuan, et al. "Duoattention: Efficient long-context llm inference with retrieval and streaming heads." *arXiv preprint arXiv:2410.10819* (2024).

[4] Fu, Yu, et al. "Not all heads matter: A head-level kv cache compression method with integrated retrieval and reasoning." *arXiv preprint arXiv:2410.19258* (2024).

**Questions:**

* Regarding the main finding — why can the locally focused and globally focused dimensions be separated by contiguous index ranges? Wouldn’t one expect them to be more interleaved or randomly distributed across dimensions?

---

> ### Author Response · Authors · 2025-11-21
> **Reply (1/4)**
>
> First, we thank the reviewer for recognizing our novel observation on head dimension heterogeneity, distinct perspective on compression scheme, and solid system implementation.
>
> > Regarding W1 on efficiency evaluation
>
> Thank you for your suggestion. We have further refined our efficiency comparison experiments by supplementing throughput measurements for each model during the decoding phase, which are recorded in the table below. As shown in the table, our method demonstrates higher throughput than other approaches during the decoding phase. We will carefully revise this part in our paper. Thank you for your review.
>
> | Throughput (token/s) | 8K      | 16K    | 32K    | 48K    | 64K    | 80k    |
> |:---------------------|:-------:|:------:|:------:|:------:|:------:|:------:|
> |***Llama 3.1-8B***    |         |        |        |        |        |        |
> | + SnapKV             | 77.49   | 40.35  | 19.73  | 16.50  | oom    | oom    |
> | + Palu               | 109.54  | 50.91  | 27.61  | 17.32  | 13.66  | oom    |
> | + FA (ours)          | 127.05  | 64.58  | 35.02  | 21.23  | 17.46  | 11.81  |
> |***Llama 3.2-3B***    |         |        |        |        |        |        |
> | + SnapKV             | 118.32  | 60.85  | 30.48  | 23.56  | oom    | oom    |
> | + Palu               | 152.75  | 70.73  | 39.35  | 29.96  | 18.65  | 14.88  |
> | + FA (ours)          | 185.20  | 93.14  | 55.25  | 33.67  | 24.16  | 20.72  |
>
> > Regarding W2 on model generalization
>
> Thank you for your question. We agree that a key measure of our method's generalizability is whether its core assumption, the heterogeneity of dimensions within attention heads, holds for different head dimension sizes (head_dim) and across various model architectures. To systematically address your question, we have conducted two sets of experiments.

---

> ### Author Response · Authors · 2025-11-21
> **Reply (2/4)**
>
> To directly examine the performance of our method on a model with a smaller head dim, we follow your suggestion and conduct a comprehensive evaluation on the Llama 3.2-1B model. The head_dim of this model (64) is smaller than that of the 8B model (128) in our original paper. As can be seen in the table, even with a significantly reduced head_dim, our method also achieves excellent performance on Llama 3.2-1B. At similar compression ratios, our method's performance on LongBench downstream tasks consistently outperforms methods such as SnapKV and PyramidKV. Therefore, our method possesses good generalization ability for models with varying head dimension sizes.
>
> |             | SD  |      |      | MD  |      |     | Sum  |      |      | ICL  |      |      | Syn |     | Code |      | Avg. |
> |:------------|:---:|:----:|:----:|:---:|:----:|:---:|:----:|:----:|:----:|:----:|:----:|:----:|:---:|:---:|:----:|:----:|:----:|
> |             | NQ  | Qsp  | MF   | HQ  | WQ   | Msq | GR   | QS   | MN   | TR   | TQ   | SS   | PC  | PR  | LCC  | Re-P |      |
> |***Llama3.2-1B***|*5.1*|*20.2*|*24.9*|*9.1*|*10.3*|*6.2*|*28.2*|*20.8*|*23.4*|*67.5*|*82.1*|*39.5*|*1.4*|*3.4*|*56.7*|*57.2*|*33.1*|
> | + SLM       | 5.8 | 11.8 | 16.1 | 6.1 | 9.5  | 3.4 | 16.5 | 19.8 | 20.1 | 50.5 | 68.4 | 37.2 | 1.3 | 4.1 | 48.1 | 48.4 | 27.1 |
> | + SnapKV    | 4.1 | 19.7 | 24.2 | 9.3 | 10.1 | 6.0 | 28.4 | 20.4 | 23.2 | 66.5 | 81.2 | 39.2 | 1.5 | 3.4 | 46.2 | 47.7 | 30.2 |
> | + PyramidKV | 4.3 | 19.6 | 24.3 | 9.1 | 10.1 | 6.0 | 28.4 | 20.5 | 23.1 | 67.0 | 81.3 | 39.3 | 1.5 | 3.4 | 46.2 | 47.7 | 30.2 |
> | + PALU      | 1.9 | 10.0 | 13.4 | 3.9 | 7.8  | 1.2 | 10.3 | 4.3  | 9.2  | 51.0 | 28.2 | 12.3 | 0.5 | 3.5 | 44.3 | 40.9 | 19.6 |
> | + FA(ours)  | 2.4 | 17.1 | 24.0 | 8.8 | 10.5 | 4.9 | 16.2 | 19.6 | 19.6 | 63.5 | 80.8 | 39.5 | 1.3 | 4.1 | 56.8 | 51.7 |**30.8**|
>
> To further alleviate concerns about architectural dependency, we also conduct validations on Qwen2.5 architecture (the Qwen2.5-1.5B and Qwen2.5-3B models). The results are as follows. As can be seen, our method continues to maintain a leading performance on the Qwen models on LongBench. This demonstrates that the effectiveness of our method is not limited to Llama architecture. Its core principles can be successfully transferred to other mainstream Transformer architectures, yielding similar performance gains.
>
> |              | SD   |      |      | MD   |      |      | Sum  |      |      | ICL  |      |      | Syn |     | Code |      | Avg. |
> |:-------------|:----:|:----:|:----:|:----:|:----:|:----:|:----:|:----:|:----:|:----:|:----:|:----:|:---:|:---:|:----:|:----:|:-----|
> |              | NQ   | Qsp  | MF   | HQ   | WQ   | Msq  | GR   | QS   | MN   | TR   | TQ   | SS   | PC  | PR  | LCC  | Re-P |      |
> |***Qwen2.5-1.5B***|*10.6*|*24.5*|*36.6*|*23.4*|*21.8*|*11.0*|*30.2*|*23.5*|*26.1*|*71.5*|*77.9*|*42.9*|*2.0*|*7.5*|*56.1*|*58.8*|*36.7*|
> | + SLM        | 12.7 | 13.6 | 26.1 | 10.2 | 14.6 | 5.3  | 21.5 | 20.4 | 23.9 | 56.5 | 56.6 | 40.5 | 1.4 | 5.8 | 48.9 | 46.9 | 28.9 |
> | + SnapKV     | 8.9  | 24.6 | 36.3 | 22.0 | 21.8 | 10.5 | 30.0 | 23.0 | 25.9 | 71.0 | 76.9 | 42.1 | 2.0 | 7.5 | 45.4 | 50.3 | 33.7 |
> | + PyramidKV  | 8.9  | 24.5 | 36.3 | 21.7 | 21.9 | 10.5 | 30.0 | 23.1 | 25.9 | 71.5 | 77.0 | 42.2 | 2.0 | 7.5 | 45.4 | 50.2 | 33.7 |
> | + Palu       | 9.1  | 19.6 | 37.5 | 10.5 | 19.8 | 4.3  | 16.8 | 12.6 | 12.4 | 70.0 | 68.3 | 22.2 | 1.6 | 3.5 | 47.2 | 46.5 | 28.4 |
> | + FA(ours)   | 6.7  | 24.2 | 34.4 | 22.9 | 21.8 | 10.3 | 21.6 | 23.0 | 25.0 | 63.5 | 76.5 | 42.0 | 2.0 | 8.5 | 55.9 | 55.4 |**34.8**|
> |***Qwen2.5-3B***|*14.3*|*29.9*|*45.9*|*38.4*|*33.5*|*20.6*|*29.7*|*24.2*|*25.2*|*69.0*|*89.5*|*45.3*|*2.0*|*8.0*|*69.3*|*65.6*|*42.7*|
> | + SLM        | 13.9 | 17.6 | 25.6 | 21.3 | 23.6 | 7.8  | 22.3 | 19.9 | 24.5 | 51.5 | 79.1 | 42.4 | 1.5 | 6.5 | 56.0 | 52.7 | 33.2 |
> | + SnapKV     | 14.5 | 30.0 | 45.4 | 37.1 | 33.6 | 19.7 | 29.2 | 23.7 | 25.0 | 69.0 | 88.1 | 45.0 | 1.5 | 8.0 | 57.3 | 55.3 | 39.5 |
> | + PyramidKV  | 14.5 | 30.0 | 45.6 | 37.1 | 33.5 | 19.7 | 29.4 | 23.8 | 25.0 | 69.0 | 88.5 | 45.0 | 1.5 | 8.0 | 57.4 | 55.2 | 39.5 |
> | + Palu       | 6.3  | 15.5 | 35.2 | 11.3 | 17.3 | 5.5  | 17.6 | 11.4 | 14.2 | 68.0 | 78.3 | 26.1 | 4.7 | 5.0 | 58.1 | 55.3 | 31.5 |
> | + FA(ours)   | 6.4  | 27.7 | 43.3 | 30.4 | 31.7 | 16.3 | 22.0 | 23.4 | 24.7 | 66.5 | 88.9 | 45.2 | 2.0 | 7.5 | 68.9 | 62.0 |**40.1**|
>
> In summary, these two new sets of experiments address your concerns regarding model generalizability. Thank you again for your valuable feedback. These suggestions have greatly enhanced the persuasiveness and completeness of our work. We will incorporate the results of these two experiments into the final version of the paper.

---

> ### Author Response · Authors · 2025-11-21
> **Reply (3/4)**
>
> > Regarding W3 on per-head analysis
>
> Thank you for your insightful question. Indeed, our method is inherently heterogeneous at the head level. Our method distinguishes between "long-context-sensitive dimensions"  and "long-context-insensitive dimensions". Since different heads have varying numbers of long-context-sensitive dimensions, our approach is necessarily heterogeneous at the head level as well.
>
> To explore this issue more deeply, we also conduct a correlational analysis to link our proposed dimensional heterogeneity with the widely discussed "head heterogeneity" method. We run a new experiment primarily comparing our method with DuoAttention[1] and HeadKV[2]. DuoAttention identifies "retrieval heads" specialized in efficiently locating and extracting information within long texts. HeadKV identifies "retrieval-reasoning heads" by analyzing attention scores as the model answers questions that require reasoning.
>
> Due to the open-source availability of these works, we conduct our experiments on the Llama 3.1-8B-Instruct, Llama 3-8B-Instruct, and Mistral-7B-Instruct-v0.2 models, for which both DuoAttention and HeadKV provide resources. Our core objective is to verify whether the attention heads that contain more of our identified "long-context-sensitive dimensions" are also considered "important" by other head-level methods, i.e., the "retrieval heads" they refer to. For each layer in the model, we defined three sets of "important heads":
> - Set A (DuoAttention): We use the attention score data provided by DuoAttention and define heads whose attention scores on  Passkey Retrieval exceed a high threshold $\gamma$ as important heads (define the size of A as K).
> - Set B (HeadKV): We utilize the code and importance score data provided by HeadKV and similarly select the Top-K heads with the highest scores to form Set B.
> - Set C (Ours): First, we count how many "long-context-sensitive dimensions" (including both K and V) identified by our method are contained within each head. Then, we select the Top-K heads with the highest count of long-context-sensitive dimensions to form Set C. The final sizes of sets A, B, and C are identical.
>
> We use the standard Jaccard similarity coefficient to measure the degree of overlap between these sets. The results are summarized as follows:
>
> | Model\Similarity        | ours vs. duo | ours vs. headkv | duo vs. headkv |
> |:-------------------------|:------------:|:---------------:|:--------------:|
> | Llama 3.1-8b-Instruct    | 0.762        | 0.755           | 0.736          |
> | Llama 3-8b-Instruct      | 0.713        | 0.725           | 0.723          |
> | Mistral-v0.2-7B-Instruct | 0.884        | 0.875           | 0.889          |
>
> From these highly consistent results, we can conclude that our method exhibits remarkably high consistency with the important head sets identified by DuoAttention and HeadKV. This demonstrates that the important heads identified by our method highly overlap with the specific functional heads discovered in prior work.
>
> To further demonstrate the fine-grained superiority of our dimension-level approach compared with the head-level approach, we also perform a new experiment using a coarser, head-level strategy on our FourierAttention. For each layer, the number of heads to preserve (t) is dynamically determined by its ratio of long-context-sensitive dimensions, `t = ceil(num of long-context-sensitive dimensions / 1024 * 8`. We then keep all dimensions for the top t heads richest in sensitive dimensions and apply our FA compression to the remaining ones as "long-context-insensitive dimensions". The results are shown below.
>
> |                 | SD   |      |      | MD  |      |     | Code |      | Avg. |
> |:----------------|:----:|:----:|:----:|:---:|:----:|:---:|:----:|:----:|:----:|
> |                 | NQ   | Qsp  | MF   | HQ  | WQ   | Msq | LCC  | Re-P |      |
> |***Llama3.2-3B***|*10.3*|*21.7*|*35.5*|*9.6*|*12.8*|*6.8*|*70.0*|*66.4*|*39.9*|
> | + FA(ours)      | 12.8 | 21.1 | 35.8 | 9.8 | 11.6 | 6.5 | 69.8 | 63.3 | 39.2 |
> | + HeadFA        | 6.2  | 19.5 | 34.3 | 9.4 | 11.6 | 6.2 | 69.2 | 61.2 | 37.6 |
>
> The drop in HeadFA reveals a deeper underlying mechanism. **The fundamental reason why a head becomes functionally important is very likely because the dimensional subspace it governs contains more dimensions responsible for processing the corresponding critical information**. This provides a more fine-grained and fundamental perspective for understanding the intrinsic principles of attention mechanisms. We will provide detailed supplementary experiments, results, and discussions on this aspect in this paper. Thank you again for your valuable suggestions.

---

> ### Author Response · Authors · 2025-11-21
> **Reply (4/4)**
>
> > Regarding Q1 on the cause of the main finding
>
> Thank you for the question. In Appendix B, we present further observations on dimension heterogeneity and explain its origin. We attribute the phenomenon to rotary position embedding (RoPE). Lower dimensions, with shorter periods, focus on local semantics and remain insensitive to noise in long-context retrieval, whereas upper dimensions, with longer periods, capture long-context semantics and are markedly affected by such noise. This pattern is consistently observed in Llama 3.1-8B, Llama 3.2-3B, and Qwen3-8B with longer contexts, as well as in Llama-3-8B when extrapolation is required. Analysis of Jamba attention, which lacks RoPE, shows no such heterogeneity. For the first 64 dimensions and the last 64 dimensions, adding noise respectively produces effects that are quite close.
>
> Regarding the reviewer’s suggestion that dimension heterogeneity be interleaved or randomly distributed across dimensions, this can be achieved by reordering the rotation angles with respect to the feature-dimension indices. We will add a concise summary of Appendix B to the main text so readers can more clearly see why the locally focused and globally focused dimensions are separated into contiguous index ranges, as asked by the reviewer. Thank you for raising this point.
>
> [1] Duoattention: Efficient Long-Context LLM Inference with Retrieval and Streaming Heads. https://arxiv.org/abs/2410.10819
>
> [2] Not All Heads Matter: A Head-Level KV Cache Compression Method with Integrated Retrieval and Reasoning. https://arxiv.org/abs/2410.19258

---

> ### Author Response · Authors · 2025-11-25
> **Looking forward to receiving your feedback**
>
> Dear Reviewer wpka,
>
> We hope this message finds you well. We truly appreciate the time and effort you’ve dedicated to reviewing my work, and we would be very grateful if you could provide feedback on our rebuttal. If you require further clarification or have any additional concerns, please do not hesitate to contact us. We are more than willing to continue communicating with you.
>
> Best wishes,
>
> The Authors

---

> ### Comment · Reviewer_wpka · 2025-11-27
> **Reply to Author's Response.**
>
> * to A1. The author provide the throughput benchmark, but the experiment details like settings need to be described. For example, SnapKV is a KV Cache eviction method. This kind of methods usually set a KV Cache budget and only keep a limited size of KV Cache, and snapKV drop tokens in prefill stage. So the Out-of-Memory of SnapKV in decoding phase need more clarify.
> * to A2. The author provide experiments that proposed method works well on smaller head dim (64) models (Llama3.2-1B). But as introduced in the paper's section 3.1:
> > the first 70 dimensions (0–69) exhibit sharp focus on short-range context, with score distributions concentrated on recent tokens, while the latter 58 dimensions (70–127) maintain a persistent bias toward initial ”sink tokens”—positional embeddings that serve as static reference points.
>
> proposed finding is a fixed pattern and seems hard to adapt to model with 64 head dim. A well introduction of how to transform the split point pattern across different head dim configurations with analysis from both experiment and theorical perspectives well strength this work.

---

> > ### Author Response · Authors · 2025-11-27
> > **Reply (1/2)**
> >
> > Thank you for your valuable feedback!
> >
> > > Regarding the question on efficiency comparison
> >
> > It is an important question. We first clarify the settings used in our efficiency experiments.
> >
> > - Palu reduces KV-cache storage by compressing each KV cache into a shorter latent representation. Palu evaluates compression ratios of 30%, 50% and 70%, and the official codebase offers only these three options. We therefore adopt the officially recommended minimum compression ratio of 70%, which is closest to ours, thus keeping the comparison fair.
> > - SnapKV uses the last 32 prompt tokens as an “observation window” to decide the token importance, applies a pooling with kernel size 7 to obtain contiguous blocks, and retains the top-2052 tokens with full KV states.
> > - FourierAttention (ours). We set preserved size N' = 1024, i.e., we compress into 1024 real-valued frequency states, while keeping a local window with 1024 tokens and an attention sink with 4 tokens, giving 1024 + 1024 + 4 = 2052 tokens, identical to SnapKV’s capacity.
> >
> > We therefore align the settings as closely as possible to guarantee fair and meaningful throughput comparisons.
> >
> > Regarding the reviewer’s query about SnapKV’s OOM. The OOM occurs during the prefill phase. SnapKV’s token-eviction criterion demands extra computation and incurs non-negligible overhead. As we note in the introduction, `while these approaches enable selective token pruning with minimal accuracy degradation, they impose prohibitive memory and latency overheads due to attention-score recalculation`.  SnapKV also states in its discussion[1] that `SnapKV’s design does not cover the processing of the prompt inference, which limits its effectiveness in scenarios where the system cannot handle prompts of extensive length`. Other study[2] also reports that methods like SnapKV and H2O still have the potential to cause OOM if not combined with chunked prefilling. Hence, SnapKV’s OOM is consistent with prior observations. The tables below include all records of our experimental results for prefill latency, reserved memory, and throughput. We will clarify these in the revised paper. Thank you for your question.
> >
> > | Peak Memory (MB) | 8K     | 16K    | 32K    | 48K    | 64K    | 80k    |
> > |:-----------------|:------:|:------:|:------:|:------:|:------:|:------:|
> > |***Llama 3.1-8B***|        |        |        |        |        |        |
> > | + SnapKV         | 26.27  | 37.44  | 59.78  | 77.29  | oom    | oom    |
> > | + Palu           | 24.56  | 32.56  | 47.31  | 62.49  | 77.65  | oom    |
> > | + FA (ours)      | 18.37  | 21.39  | 27.42  | 33.51  | 39.61  | 45.72  |
> > |***Llama 3.2-3B***|        |        |        |        |        |        |
> > | + SnapKV         | 14.69  | 26.15  | 40.46  | 57.65  | oom    | oom    |
> > | + Palu           | 14.85  | 21.92  | 36.05  | 49.91  | 63.95  | 71.46  |
> > | + FA (ours)      | 9.20   | 12.08  | 17.03  | 23.57  | 29.36  | 35.15  |
> >
> > | Prefill Latency (s) | 8K    | 16K   | 32K   | 48K   | 64K   | 80k   |
> > |:--------------------|:-----:|:-----:|:-----:|:-----:|:-----:|:-----:|
> > |***Llama 3.1-8B***   |       |       |       |       |       |       |
> > | + SnapKV            | 0.29  | 0.66  | 1.65  | 3.06  | oom   | oom   |
> > | + Palu              | 0.31  | 0.69  | 1.81  | 3.33  | 5.23  | oom   |
> > | + FA (ours)         | 0.37  | 0.76  | 1.80  | 3.30  | 5.04  | 7.29  |
> > |***Llama 3.2-3B***   |       |       |       |       |       |       |
> > | + SnapKV            | 0.15  | 0.36  | 0.93  | 1.75  | oom   | oom   |
> > | + Palu              | 0.16  | 0.38  | 0.99  | 1.88  | 3.00  | 4.32  |
> > | + FA (ours)         | 0.21  | 0.44  | 1.03  | 1.95  | 2.99  | 4.44  |
> >
> > | Throughput (token/s) | 8K      | 16K    | 32K    | 48K    | 64K    | 80k    |
> > |:---------------------|:-------:|:------:|:------:|:------:|:------:|:------:|
> > |***Llama 3.1-8B***    |         |        |        |        |        |        |
> > | + SnapKV             | 77.49   | 40.35  | 19.73  | 16.50  | oom    | oom    |
> > | + Palu               | 109.54  | 50.91  | 27.61  | 17.32  | 13.66  | oom    |
> > | + FA (ours)          | 127.05  | 64.58  | 35.02  | 21.23  | 17.46  | 11.81  |
> > |***Llama 3.2-3B***    |         |        |        |        |        |        |
> > | + SnapKV             | 118.32  | 60.85  | 30.48  | 23.56  | oom    | oom    |
> > | + Palu               | 152.75  | 70.73  | 39.35  | 29.96  | 18.65  | 14.88  |
> > | + FA (ours)          | 185.20  | 93.14  | 55.25  | 33.67  | 24.16  | 20.72  |

---

> > > ### Comment · Reviewer_wpka · 2025-11-28
> > > **Acknowledge to Author's Response**
> > >
> > > * About efficiency. Thanks the author's further experiments and analysis. The settings is fair.
> > > * About dimension. I did a heatmap visualization of RoPE(x)-x on different configurations. As author's response and reference [3], the perturbated dimensions increase as positions and likely meet a upper bound around ~60% head dimensions. Maybe a ratio setting rather than a fix pattern in methodology can make proposed work more promising.

---

> > > > ### Comment · Reviewer_wpka · 2025-11-28
> > > > **About the Rebuttal**
> > > >
> > > > The author's response help me better understanding the proposed work and adressed my concern. I'm will to update my rating from 4 to 6 if the system allow later.

---

> > > > > ### Author Response · Authors · 2025-12-02
> > > > > **Reply**
> > > > >
> > > > > Thank you for recognizing the fairness of our efficiency experiments and acknowledging our response.
> > > > >
> > > > > Regarding the feedback on the dimension, we use the critical dimension as the boundary between the upper and lower dimensions. As already cited, dimensions before and after this point have been exposed to complete and incomplete positional information and thus exhibit distinct extrapolation behaviours. Our work further shows that they also perform significantly differently on long-context tasks within the supported context length. When the pre-training hyperparameters, including head dimension d, are fixed, the critical dimension d_extra is determined by the formula below.
> > > > >
> > > > > $$d_\text{extra}=2\left\lceil\frac{d}{2}\log_\beta{\frac{T_\text{train}}{2\pi}}\right\rceil$$
> > > > >
> > > > > When the head dimension d is not fixed, the ratio of critical dimension d_extra over d is determined by the pre-training rotary base and context length.
> > > > >
> > > > > $$\frac{d_\text{extra}}{d}\approx\log_\beta{\frac{T_\text{train}}{2\pi}}$$
> > > > >
> > > > > For the models studied here, this ratio is approximately 60%, but it varies slightly with different initial bases and context lengths, exhibiting adaptive behaviour.
> > > > >
> > > > > |                 | d_extra/d |
> > > > > |:---------------:|:---------:|
> > > > > | Llama 3/3.1/3.2 | 0.5466    |
> > > > > | Qwen 2/2.5/3    | 0.7035    |

---

> > ### Author Response · Authors · 2025-11-27
> > **Reply (2/2)**
> >
> > > Regarding the question on dimension heterogeneity
> >
> > Thank you for this detailed question. Appendix B explains that for Llama 3.1-8B with a head dimension of 128, we adopt a 70-58 split because 70 is the critical dimension for RoPE-based extrapolation[3], the number of dimensions that have observed a full sinusoidal period during pre-training. Dimensions beyond 70 have never seen a complete period, retain stronger monotonicity, and therefore tend to encode long-context semantics. In any RoPE-based model, regardless of feature dimension, the period of the lowest dimension ($2\pi$) is always shorter than the pre-training length, whereas the period of the highest dimension (almost $2\pi\cdot10000$) is always longer. Hence, the critical dimension and heterogeneous pattern always persist and, of course, still hold for Llama 3.2-1B with a head dimension of 64. We will add this clarification and visualized verification to the paper and thank the reviewer for this suggestion.
> >
> > [1] SnapKV: LLM Knows What You are Looking for Before Generation https://arxiv.org/abs/2404.14469
> >
> > [2] Memorize step by step: Efficient long-context prefilling with incremental memory and decremental chunk https://aclanthology.org/2024.emnlp-main.1169/
> >
> > [3] Scaling Laws of RoPE-based Extrapolation https://arxiv.org/abs/2310.05209

---

### Official Review · Reviewer_iCAS · 2025-11-01

**Soundness:** 3
**Presentation:** 3
**Contribution:** 3
**Rating:** 8
**Confidence:** 3

**Summary:**

This paper proposes FourierAttention, an online compression method for the KV cache. FourierAttention uses Hippo-FourierT to compress the KV cache, significantly reducing the memory overhead of the KV cache while maintaining better performance compared to baseline methods.

**Strengths:**

+ This paper discovers that the different dimensions of Q and K in attention computation play different roles. Initially, this finding seemed counterintuitive to me, as I typically assumed that different dimensions were homogeneous—this was because I had overlooked the effect of ROPE. The paper innovatively leverages this insight by applying different compression strategies to different dimensional ranges, achieving better compression efficiency.

+ The paper also implements the proposed method's corresponding FlashFourierAttention operator, which is an important contribution to the open-source community.

+ The experiments provide thorough comparisons in terms of performance and latency, effectively demonstrating the superiority of the proposed method.

**Weaknesses:**

+ The paper only conducts experiments on LLaMA. It would be better to include comparisons with other open-source models as well.

**Questions:**

See above

---

> ### Author Response · Authors · 2025-11-21
> **Reply**
>
> First, we appreciate the reviewer for recognizing our discovery on dimension heterogeneity, thorough comparisons in terms of performance and latency, as well as our custom Triton operator.
>
> > Regarding W1 on the evaluation of other LLMs
>
> Thank you for your question. To comprehensively validate the generalizability of our method, we have supplemented our original experiments on the Llama series models with evaluations on the Qwen2.5 architecture (the Qwen2.5-1.5B and Qwen2.5-3B models). As with the Llama models, we select the LongBench benchmark for long-context understanding to conduct the evaluation, using an experimental setup similar to that for the Llama models. The experiments show that our method continues to demonstrate strong competitiveness on the Qwen2.5 models. The detailed results are shown in the table below.
>
> |              | SD   |      |      | MD   |      |      | Sum  |      |      | ICL  |      |      | Syn |     | Code |      | Avg. |
> |:-------------|:----:|:----:|:----:|:----:|:----:|:----:|:----:|:----:|:----:|:----:|:----:|:----:|:---:|:---:|:----:|:----:|:-----|
> |              | NQ   | Qsp  | MF   | HQ   | WQ   | Msq  | GR   | QS   | MN   | TR   | TQ   | SS   | PC  | PR  | LCC  | Re-P |      |
> |***Qwen2.5-1.5B***|*10.6*|*24.5*|*36.6*|*23.4*|*21.8*|*11.0*|*30.2*|*23.5*|*26.1*|*71.5*|*77.9*|*42.9*|*2.0*|*7.5*|*56.1*|*58.8*|*36.7*|
> | + SLM        | 12.7 | 13.6 | 26.1 | 10.2 | 14.6 | 5.3  | 21.5 | 20.4 | 23.9 | 56.5 | 56.6 | 40.5 | 1.4 | 5.8 | 48.9 | 46.9 | 28.9 |
> | + SnapKV     | 8.9  | 24.6 | 36.3 | 22.0 | 21.8 | 10.5 | 30.0 | 23.0 | 25.9 | 71.0 | 76.9 | 42.1 | 2.0 | 7.5 | 45.4 | 50.3 | 33.7 |
> | + PyramidKV  | 8.9  | 24.5 | 36.3 | 21.7 | 21.9 | 10.5 | 30.0 | 23.1 | 25.9 | 71.5 | 77.0 | 42.2 | 2.0 | 7.5 | 45.4 | 50.2 | 33.7 |
> | + Palu       | 9.1  | 19.6 | 37.5 | 10.5 | 19.8 | 4.3  | 16.8 | 12.6 | 12.4 | 70.0 | 68.3 | 22.2 | 1.6 | 3.5 | 47.2 | 46.5 | 28.4 |
> | + FA(ours)   | 6.7  | 24.2 | 34.4 | 22.9 | 21.8 | 10.3 | 21.6 | 23.0 | 25.0 | 63.5 | 76.5 | 42.0 | 2.0 | 8.5 | 55.9 | 55.4 |**34.8**|
> |***Qwen2.5-3B***|*14.3*|*29.9*|*45.9*|*38.4*|*33.5*|*20.6*|*29.7*|*24.2*|*25.2*|*69.0*|*89.5*|*45.3*|*2.0*|*8.0*|*69.3*|*65.6*|*42.7*|
> | + SLM        | 13.9 | 17.6 | 25.6 | 21.3 | 23.6 | 7.8  | 22.3 | 19.9 | 24.5 | 51.5 | 79.1 | 42.4 | 1.5 | 6.5 | 56.0 | 52.7 | 33.2 |
> | + SnapKV     | 14.5 | 30.0 | 45.4 | 37.1 | 33.6 | 19.7 | 29.2 | 23.7 | 25.0 | 69.0 | 88.1 | 45.0 | 1.5 | 8.0 | 57.3 | 55.3 | 39.5 |
> | + PyramidKV  | 14.5 | 30.0 | 45.6 | 37.1 | 33.5 | 19.7 | 29.4 | 23.8 | 25.0 | 69.0 | 88.5 | 45.0 | 1.5 | 8.0 | 57.4 | 55.2 | 39.5 |
> | + Palu       | 6.3  | 15.5 | 35.2 | 11.3 | 17.3 | 5.5  | 17.6 | 11.4 | 14.2 | 68.0 | 78.3 | 26.1 | 4.7 | 5.0 | 58.1 | 55.3 | 31.5 |
> | + FA(ours)   | 6.4  | 27.7 | 43.3 | 30.4 | 31.7 | 16.3 | 22.0 | 23.4 | 24.7 | 66.5 | 88.9 | 45.2 | 2.0 | 7.5 | 68.9 | 62.0 |**40.1**|
>
> Compared to various KV Cache compression methods such as SnapKV, our method achieves performance improvements on LongBench, which is consistent with the trend we observed on the Llama models.
>
> Furthermore, to demonstrate the effectiveness of our method with different head dimension sizes (head_dim), we also conduct the same experiments on the Llama 3.2-1B model (which has a head dimension of 64, a significant difference from the 128 of the Llama 3.2-3B model). The results are as follows:
>
> |             | SD  |      |      | MD  |      |     | Sum  |      |      | ICL  |      |      | Syn |     | Code |      | Avg. |
> |:------------|:---:|:----:|:----:|:---:|:----:|:---:|:----:|:----:|:----:|:----:|:----:|:----:|:---:|:---:|:----:|:----:|:----:|
> |             | NQ  | Qsp  | MF   | HQ  | WQ   | Msq | GR   | QS   | MN   | TR   | TQ   | SS   | PC  | PR  | LCC  | Re-P |      |
> |***Llama3.2-1B***|*5.1*|*20.2*|*24.9*|*9.1*|*10.3*|*6.2*|*28.2*|*20.8*|*23.4*|*67.5*|*82.1*|*39.5*|*1.4*|*3.4*|*56.7*|*57.2*|*33.1*|
> | + SLM       | 5.8 | 11.8 | 16.1 | 6.1 | 9.5  | 3.4 | 16.5 | 19.8 | 20.1 | 50.5 | 68.4 | 37.2 | 1.3 | 4.1 | 48.1 | 48.4 | 27.1 |
> | + SnapKV    | 4.1 | 19.7 | 24.2 | 9.3 | 10.1 | 6.0 | 28.4 | 20.4 | 23.2 | 66.5 | 81.2 | 39.2 | 1.5 | 3.4 | 46.2 | 47.7 | 30.2 |
> | + PyramidKV | 4.3 | 19.6 | 24.3 | 9.1 | 10.1 | 6.0 | 28.4 | 20.5 | 23.1 | 67.0 | 81.3 | 39.3 | 1.5 | 3.4 | 46.2 | 47.7 | 30.2 |
> | + Palu      | 1.9 | 10.0 | 13.4 | 3.9 | 7.8  | 1.2 | 10.3 | 4.3  | 9.2  | 51.0 | 28.2 | 12.3 | 0.5 | 3.5 | 44.3 | 40.9 | 19.6 |
> | + FA(ours)  | 2.4 | 17.1 | 24.0 | 8.8 | 10.5 | 4.9 | 16.2 | 19.6 | 19.6 | 63.5 | 80.8 | 39.5 | 1.3 | 4.1 | 56.8 | 51.7 |**30.8**|
>
> The experiments show that our method can achieve consistently effective performance on models with different architectures and head dimensions, which strongly proves that our method possesses good cross-architecture generalization ability. Thank you again for your suggestion. We will organize these detailed experimental results and analysis and incorporate them into the paper.

---

### Official Review · Reviewer_sDbS · 2025-11-03

**Soundness:** 2
**Presentation:** 3
**Contribution:** 3
**Rating:** 4
**Confidence:** 4

**Summary:**

This paper proposes FourierAttention, a training-free KV cache compression method. The idea is that KV cache tokens have long-context-insensitive and long-context sensitive channels -- the former can be projected onto translated Fourier bases, while the latter can be ;eft uncompressed. Authors also propose a Triton kernel for on-the-fly reconstruction of the tokens. Experiments on long-context benchmarks with LLaMA-3.* models show improvements over baseline KV cache compression methods like SnapKV and PyramidKV. One concern I have is that results in Tab. 1 often show very marginal improvements, and results are often worse than those of SnapKV and PyramidKV -- are those statistically significant? Also, authors only experiment with Llama-3.1-8B and Llama-3.2-3B -- what happens on models not in the Llama3 family (e.g. any of the recent Qwens)? On the other hand, the baselines are very competitive (according to e.g. KVPress, https://github.com/NVIDIA/kvpress) so any improvements on those is more than welcome.

Tiny typo: line 189 -- "We denode"

**Strengths:**

- Interesting analysis about how latent dimensions can be characterised into long-context-sensitive/insensitive (also from a mechinterp point of view)
- Strong results compared with competitive baselines like SnapKV, PyramidKV, and Palu at comparable budgets on LongBench and NIAH
- Interesting custom FlashFourierAttention kernel

**Weaknesses:**

- Absolute gains are *very* modest -- are they statistically significant?
- I was not able to find quantitative results on latency, please let me know if I missed those
- Experiments only on Llama3-based backbones

**Questions:**

- What happens on other families on models?
- Are results statistically significant?
- How sensitive are the results to errors in identifying the two types of latent dimensions?

---

> ### Author Response · Authors · 2025-11-21
> **Reply (1/3)**
>
> First, we appreciate the reviewer for recognizing our analysis on dimension heterogeneity, superiority in long-context performance over other cache optimization methods, as well as our custom Triton kernel.
>
> > Regarding W3 and Q1 on evaluation on other LLMs
>
> Thank you for your question. To comprehensively validate the generalizability of our method, we have supplemented our original experiments on the Llama series models with evaluations on Qwen2.5 architecture (the Qwen2.5-1.5B and Qwen2.5-3B models). As with the Llama models, we select the LongBench benchmark for long-context understanding to conduct the evaluation, using an experimental setup similar to that for the Llama models. The experiments show that our method continues to demonstrate strong competitiveness on the Qwen2.5 models. The detailed results are shown in the table below. Compared to various KV Cache compression methods such as SnapKV, our method achieves performance improvements on LongBench, which is consistent with the trend we observed on the Llama models.
>
> |              | SD   |      |      | MD   |      |      | Sum  |      |      | ICL  |      |      | Syn |     | Code |      | Avg. |
> |:-------------|:----:|:----:|:----:|:----:|:----:|:----:|:----:|:----:|:----:|:----:|:----:|:----:|:---:|:---:|:----:|:----:|:-----|
> |              | NQ   | Qsp  | MF   | HQ   | WQ   | Msq  | GR   | QS   | MN   | TR   | TQ   | SS   | PC  | PR  | LCC  | Re-P |      |
> |***Qwen2.5-1.5B***|*10.6*|*24.5*|*36.6*|*23.4*|*21.8*|*11.0*|*30.2*|*23.5*|*26.1*|*71.5*|*77.9*|*42.9*|*2.0*|*7.5*|*56.1*|*58.8*|*36.7*|
> | + SLM        | 12.7 | 13.6 | 26.1 | 10.2 | 14.6 | 5.3  | 21.5 | 20.4 | 23.9 | 56.5 | 56.6 | 40.5 | 1.4 | 5.8 | 48.9 | 46.9 | 28.9 |
> | + SnapKV     | 8.9  | 24.6 | 36.3 | 22.0 | 21.8 | 10.5 | 30.0 | 23.0 | 25.9 | 71.0 | 76.9 | 42.1 | 2.0 | 7.5 | 45.4 | 50.3 | 33.7 |
> | + PyramidKV  | 8.9  | 24.5 | 36.3 | 21.7 | 21.9 | 10.5 | 30.0 | 23.1 | 25.9 | 71.5 | 77.0 | 42.2 | 2.0 | 7.5 | 45.4 | 50.2 | 33.7 |
> | + Palu       | 9.1  | 19.6 | 37.5 | 10.5 | 19.8 | 4.3  | 16.8 | 12.6 | 12.4 | 70.0 | 68.3 | 22.2 | 1.6 | 3.5 | 47.2 | 46.5 | 28.4 |
> | + FA(ours)   | 6.7  | 24.2 | 34.4 | 22.9 | 21.8 | 10.3 | 21.6 | 23.0 | 25.0 | 63.5 | 76.5 | 42.0 | 2.0 | 8.5 | 55.9 | 55.4 |**34.8**|
> |***Qwen2.5-3B***|*14.3*|*29.9*|*45.9*|*38.4*|*33.5*|*20.6*|*29.7*|*24.2*|*25.2*|*69.0*|*89.5*|*45.3*|*2.0*|*8.0*|*69.3*|*65.6*|*42.7*|
> | + SLM        | 13.9 | 17.6 | 25.6 | 21.3 | 23.6 | 7.8  | 22.3 | 19.9 | 24.5 | 51.5 | 79.1 | 42.4 | 1.5 | 6.5 | 56.0 | 52.7 | 33.2 |
> | + SnapKV     | 14.5 | 30.0 | 45.4 | 37.1 | 33.6 | 19.7 | 29.2 | 23.7 | 25.0 | 69.0 | 88.1 | 45.0 | 1.5 | 8.0 | 57.3 | 55.3 | 39.5 |
> | + PyramidKV  | 14.5 | 30.0 | 45.6 | 37.1 | 33.5 | 19.7 | 29.4 | 23.8 | 25.0 | 69.0 | 88.5 | 45.0 | 1.5 | 8.0 | 57.4 | 55.2 | 39.5 |
> | + Palu       | 6.3  | 15.5 | 35.2 | 11.3 | 17.3 | 5.5  | 17.6 | 11.4 | 14.2 | 68.0 | 78.3 | 26.1 | 4.7 | 5.0 | 58.1 | 55.3 | 31.5 |
> | + FA(ours)   | 6.4  | 27.7 | 43.3 | 30.4 | 31.7 | 16.3 | 22.0 | 23.4 | 24.7 | 66.5 | 88.9 | 45.2 | 2.0 | 7.5 | 68.9 | 62.0 |**40.1**|
>
> Furthermore, to demonstrate the effectiveness of our method with different head dimension sizes (head_dim), we also conduct the same experiments on the Llama 3.2-1B model (which has a head dimension of 64, a significant difference from the 128 of the Llama 3.2-3B model). The results are as follows:
>
> |             | SD  |      |      | MD  |      |     | Sum  |      |      | ICL  |      |      | Syn |     | Code |      | Avg. |
> |:------------|:---:|:----:|:----:|:---:|:----:|:---:|:----:|:----:|:----:|:----:|:----:|:----:|:---:|:---:|:----:|:----:|:----:|
> |             | NQ  | Qsp  | MF   | HQ  | WQ   | Msq | GR   | QS   | MN   | TR   | TQ   | SS   | PC  | PR  | LCC  | Re-P |      |
> |***Llama3.2-1B***|*5.1*|*20.2*|*24.9*|*9.1*|*10.3*|*6.2*|*28.2*|*20.8*|*23.4*|*67.5*|*82.1*|*39.5*|*1.4*|*3.4*|*56.7*|*57.2*|*33.1*|
> | + SLM       | 5.8 | 11.8 | 16.1 | 6.1 | 9.5  | 3.4 | 16.5 | 19.8 | 20.1 | 50.5 | 68.4 | 37.2 | 1.3 | 4.1 | 48.1 | 48.4 | 27.1 |
> | + SnapKV    | 4.1 | 19.7 | 24.2 | 9.3 | 10.1 | 6.0 | 28.4 | 20.4 | 23.2 | 66.5 | 81.2 | 39.2 | 1.5 | 3.4 | 46.2 | 47.7 | 30.2 |
> | + PyramidKV | 4.3 | 19.6 | 24.3 | 9.1 | 10.1 | 6.0 | 28.4 | 20.5 | 23.1 | 67.0 | 81.3 | 39.3 | 1.5 | 3.4 | 46.2 | 47.7 | 30.2 |
> | + Palu      | 1.9 | 10.0 | 13.4 | 3.9 | 7.8  | 1.2 | 10.3 | 4.3  | 9.2  | 51.0 | 28.2 | 12.3 | 0.5 | 3.5 | 44.3 | 40.9 | 19.6 |
> | + FA(ours)  | 2.4 | 17.1 | 24.0 | 8.8 | 10.5 | 4.9 | 16.2 | 19.6 | 19.6 | 63.5 | 80.8 | 39.5 | 1.3 | 4.1 | 56.8 | 51.7 |**30.8**|
>
> The experiments show that our method can achieve consistently effective performance on models with different architectures and head dimensions, which strongly proves that our method possesses good cross-architecture generalization ability. Thank you again for your suggestion. We will organize these detailed experimental results and analysis and incorporate them into the paper.

---

> ### Author Response · Authors · 2025-11-21
> **Reply (2/3)**
>
> > Regarding W2 on quantitative results of efficiency comparison
>
> Thank you for raising this good question. The relevant visualization results are presented in Figure 7 of our paper. The table below also includes detailed records of our experimental results for prefill latency and reserved memory. We have also supplemented experiments on throughput during the decoding process for each model at various context lengths, which are presented below. As shown in the table, we have achieved a latency close to that of existing efficient approaches such as Palu and SnapKV in long contexts ranging from 16k to 80k, while the memory footprint is notably lower than other methods across all context lengths. In addition, our method demonstrates higher throughput than other approaches during the decoding phase. We will carefully revise this part in our paper. Thank you for your review.
>
> | Peak Memory (MB) | 8K     | 16K    | 32K    | 48K    | 64K    | 80k    |
> |:-----------------|:------:|:------:|:------:|:------:|:------:|:------:|
> |***Llama 3.1-8B***|        |        |        |        |        |        |
> | + SnapKV         | 26.27  | 37.44  | 59.78  | 77.29  | oom    | oom    |
> | + Palu           | 24.56  | 32.56  | 47.31  | 62.49  | 77.65  | oom    |
> | + FA (ours)      | 18.37  | 21.39  | 27.42  | 33.51  | 39.61  | 45.72  |
> |***Llama 3.2-3B***|        |        |        |        |        |        |
> | + SnapKV         | 14.69  | 26.15  | 40.46  | 57.65  | oom    | oom    |
> | + Palu           | 14.85  | 21.92  | 36.05  | 49.91  | 63.95  | 71.46  |
> | + FA (ours)      | 9.20   | 12.08  | 17.03  | 23.57  | 29.36  | 35.15  |
>
> | Prefill Latency (s) | 8K    | 16K   | 32K   | 48K   | 64K   | 80k   |
> |:--------------------|:-----:|:-----:|:-----:|:-----:|:-----:|:-----:|
> |***Llama 3.1-8B***   |       |       |       |       |       |       |
> | + SnapKV            | 0.29  | 0.66  | 1.65  | 3.06  | oom   | oom   |
> | + Palu              | 0.31  | 0.69  | 1.81  | 3.33  | 5.23  | oom   |
> | + FA (ours)         | 0.37  | 0.76  | 1.80  | 3.30  | 5.04  | 7.29  |
> |***Llama 3.2-3B***   |       |       |       |       |       |       |
> | + SnapKV            | 0.15  | 0.36  | 0.93  | 1.75  | oom   | oom   |
> | + Palu              | 0.16  | 0.38  | 0.99  | 1.88  | 3.00  | 4.32  |
> | + FA (ours)         | 0.21  | 0.44  | 1.03  | 1.95  | 2.99  | 4.44  |
>
> | Throughput (token/s) | 8K      | 16K    | 32K    | 48K    | 64K    | 80k    |
> |:---------------------|:-------:|:------:|:------:|:------:|:------:|:------:|
> |***Llama 3.1-8B***    |         |        |        |        |        |        |
> | + SnapKV             | 77.49   | 40.35  | 19.73  | 16.50  | oom    | oom    |
> | + Palu               | 109.54  | 50.91  | 27.61  | 17.32  | 13.66  | oom    |
> | + FA (ours)          | 127.05  | 64.58  | 35.02  | 21.23  | 17.46  | 11.81  |
> |***Llama 3.2-3B***    |         |        |        |        |        |        |
> | + SnapKV             | 118.32  | 60.85  | 30.48  | 23.56  | oom    | oom    |
> | + Palu               | 152.75  | 70.73  | 39.35  | 29.96  | 18.65  | 14.88  |
> | + FA (ours)          | 185.20  | 93.14  | 55.25  | 33.67  | 24.16  | 20.72  |
>
> > Regarding W1 and Q2 on statistical significance
>
> Thank you for your question. While the improvement of our method over other approaches on downstream tasks is modest, we achieve scores closest to full attention among all methods, while also having the lowest memory overhead. This demonstrates the effectiveness of our approach to optimizing memory from the perspective of heterogeneous features. Furthermore, as mentioned earlier, we have achieved consistent improvements across multiple LLMs, which statistically demonstrates that our advantage is sufficiently consistent.

---

> ### Author Response · Authors · 2025-11-21
> **Reply (3/3)**
>
> > Regarding W3 on the sensitivity of dimension selection
>
> Thank you for this insightful question, which prompts us to provide a more thorough ablation study and comparative analysis for our dimension selection scheme.
>
> In the paper, we have already presented several comparative experiments that indirectly validate the effectiveness of our dimension selection scheme. For example, in Table 2, for the inverted-pyramid, asymmetric compression scheme mentioned in the paper, we test the following alternative strategies on the Llama3.2-3B model for the ruler 4k under an equivalent compression ratio: replacing it with a uniform compression scheme (uniform), swapping the compression ratios for K and V (KV inv.), and reversing the inverted pyramid of layer-wise compression (layer inv.). The final results all show a consistent drop in performance, which indicates that our compression scheme is sensitive to dimension selection and that an appropriate selection scheme is key to achieving optimal performance.
>
> Furthermore, to answer this question more clearly, we also conduct a new perturbation experiment to simulate small-scale, random errors in dimension identification. First, following our original method, we identify the set of "non-critical dimensions" to be compressed for each layer of the model. Then, we simulate an "identification error" scenario: in each layer, we randomly remove 10% of these "non-critical dimensions" and replace them with an equal number of randomly selected "critical dimensions" from the same layer. The results show that with this 10% dimension identification error rate, the model's final performance shows severe degradation.
>
> To sum up, this series of experiments strongly demonstrates the high importance of our dimension selection scheme. The leading performance of our method is built upon the accurate identification and differential treatment based on heterogeneous features of head dimensions. Thank you again for your question. We plan to add this new perturbation experiment to the paper to further strengthen our claims.
>
> |               | SK1    | SK2    | SK3    | MK1   | MK2    | MK3    | MV     | MQ    | CWE   | FWE   | VT    | SQ    | HP    | Avg   |
> |:--------------|:------:|:------:|:------:|:-----:|:------:|:------:|:------:|:-----:|:-----:|:-----:|:-----:|:-----:|:-----:|:-----:|
> |***Llama 3.2-3B***|*100.0* |*100.0* |*100.0* |*99.0* |*100.0* |*99.0 * |*100.0* |*99.8* |*60.0* |*89.7* |*97.0* |*77.0* |*53.0* |*90.3* |
> | + FA(ours)    | 100.0  | 100.0  | 98.0   | 99.0  | 99.0   | 100.0  | 96.0   | 97.8  | 57.9  | 80.7  | 76.6  | 81.0  | 51.0  |**87.5**|
> | + uniform     | 100.0  | 100.0  | 99.0   | 99.0  | 99.0   | 98.0   | 93.0   | 98.8  | 59.7  | 77.3  | 78.2  | 80.0  | 50.0  | 87.1  |
> | + KV inv.     | 100.0  | 100.0  | 100.0  | 99.0  | 99.0   | 100.0  | 93.5   | 98.8  | 54.3  | 79.7  | 72.6  | 80.0  | 53.0  | 86.9  |
> | + Layer inv.  | 100.0  | 98.0   | 88.0   | 98.0  | 97.0   | 94.0   | 80.0   | 96.8  | 41.3  | 69.3  | 61.4  | 80.0  | 50.0  | 81.1  |
> | + random 10%  | 90.0   | 87.0   | 35.0   | 82.0  | 69.0   | 50.0   | 52.8   | 63.3  | 5.2   | 55.0  | 28.8  | 81.0  | 50.0  | 57.6  |

---

> ### Author Response · Authors · 2025-11-25
> **Looking forward to receiving your feedback**
>
> Dear Reviewer sDbS,
>
> We hope this message finds you well. We truly appreciate the time and effort you’ve dedicated to reviewing my work, and we would be very grateful if you could provide feedback on our rebuttal. If you require further clarification or have any additional concerns, please do not hesitate to contact us. We are more than willing to continue communicating with you.
>
> Best wishes,
>
> The Authors

---

### Official Review · Reviewer_smj6 · 2025-11-07

**Soundness:** 2
**Presentation:** 3
**Contribution:** 2
**Rating:** 2
**Confidence:** 4

**Summary:**

This paper proposes a training-free KV cache compression method that exploits heterogeneous roles across transformer head dimensions. The authors observe that lower dimensions capture local context while upper dimensions capture long-range dependencies, and compress the former using Fourier basis functions from the HiPPO framework. A custom Triton kernel (FlashFourierAttention) is implemented for efficient deployment. The observation and the custom kernel are nice, but the evaluation is not comprehensive enough. Both the choice of the baselines (compression vs quantization) and the range of performed experiments and models selected are not enough. Also, the paper would benefit from a better framing of the method with respect to the literature about KV Cache compression.

**Strengths:**

- The empirical finding that different head dimensions serve distinct roles (local vs. global context) is interesting and well-validated through ablation studies
- Leveraging the HiPPO framework provides mathematical rigor, and the adaptation from complex to real-valued representations is sensible for practical implementation.
- The custom Triton kernel shows practical engineering effort and is very nice.

**Weaknesses:**

- (General) The title and abstract claim this is "KV cache compression," but the actual mechanism is better described as lossy approximation or dimensionality reduction. For this reason, while comparing to Palu (dimensionality reduction) makes sense, comparing to SnapKV and PyramidKV is kind of strange as these perform KV Cache eviction in a different setting. The authors should (a) discuss the differences between KV Cache compression, quantization, reduction in a clear way and (b) include quantization in the baselines.
- (Method) The observation about dimension heterogeneity is a bit over-claimed and not entirely new, as prior work has noted similar patterns. This is cited by the authors themselves.
- In Table 1. The comparison shows that the methods have been tested with different compression ratios. Why not use the same compression ratio for fair comparison?
- In Table 1 and in general across the paper, for a fair and comprehensive evaluation one should consider a range of compression rations for all methods instead of a specific one.
- The choice of L-init=4, L-local=1024, N=512 appears arbitrary with no ablation study or principled justification. How sensitive is performance to these choices?
- The method is only evaluated on Llama models. This strongly hinders the generalization of the method to different architectures.
- I am not sure I understood how is the dimension selection actually performed? The authors say "we directly compress and decompress all KV caches, prioritizing dimensions with smaller mean-squared error" but doesn't specify the algorithm. Is this done once offline or adaptively?

**Questions:**

- Could you frame the contribution wrt KV Cache compression, Quantization and Low Rank reduction.
- Could you provide exact numerical results for latency and memory consumption rather than just plots?
- Why aren't other architectures included in the evaluation ? Could you include them ?
- Could you provide results in Table1 for different compression ratios ?

---

> ### Author Response · Authors · 2025-11-21
> **Reply (1/5)**
>
> First, we thank the reviewer for recognizing our well-validated dimension-heterogeneity finding, mathematically rigorous compression scheme, and the custom Triton kernel.
>
> > Regarding W1 and Q1 on cache compression
>
> Thank you for your question. **Regarding the definition of "KV Cache compression"**, our understanding is based on existing analyses and surveys[1-3]. KV Cache compression is systematically categorized into five different aspects: **Token, Layer, Head, Feature Dimension, and Number of Bytes** (quantization). Any method that performs reduction along any of these aspects can be called KV Cache compression.
>
> Our method, FourierAttention, performs compression primarily on the **Token dimension**[4-7]:
> - First, we conduct an offline analysis by compressing and decompressing each feature dimension along the token direction using Fourier transform. Based on reconstruction loss, we distinguish between "long context insensitive dimensions" and "long context sensitive dimensions."
> - Then, for the identified "long context insensitive dimensions," we compress them to a fixed length along the Token dimension using Fourier transform, thereby achieving compression of the KV Cache.
> - Therefore, although our method involves a feature dimension selection step, the fundamental compressing operation occurs on the Token dimension. Conclusively, our work is indeed a form of KV Cache compression and, more specifically, a novel method within the important sub-branch of token-dimension compression.
>
> **Regarding Comparison**
> - Therefore, it is natural for us to compare our method with classic methods in token compression like StreamingLLM[4], SnapKV[5], and PyramidKV[6]. This ensures our evaluation is conducted on a fair and level playing field.
> - Meanwhile, our method innovatively introduces a heterogeneous treatment of feature dimensions, so we also compare it with Palu[8], a representative work in dimensionality reduction, to comprehensively demonstrate the overall advantages of our method.
>
> **Regarding Quantization**
>
> We strongly agree that it is an important direction in KV Cache compression. However, we note that the token-dimension compression works we primarily compare against, such as SnapKV and PyramidKV, also did not include quantization methods as their main baselines. In fact, our method and quantization are two completely orthogonal technical dimensions, and they can work in synergy: quantization can be applied to the KV Cache that has already been compressed by our FourierAttention method to achieve further compression. The potential of this combination has been mentioned in early works (e.g., FastGen[7]). We will explore this combination more deeply in future work. Thank you again for your question.
>
> > Regarding W2 on novelty of dimension heterogeneity
>
> Thank you for your question. Earlier work does observe that certain dimensions matter more in long contexts[9-10], yet those observations remain a case study or speculation. Moreover, our work makes at least the following distinctive contributions, none of which are mentioned in the papers above:
> 1. We formally demonstrate that, in RoPE-based attention, lower dimensions prioritize local context and correspond to the local branch in StreamingLM[4], while the upper ones capture long-range dependencies and correspond to the global branch, with a clear critical dimension[11] as the boundary.
> 2. We introduce noise experiments that show injecting noise into the two components has markedly different impacts on long-context tasks. This observation motivates FourierAttention, which compresses the majority of long-context-insensitive dimensions to a fixed length. The method achieves a better efficiency and performance, and we verify that these insensitive dimensions are predominantly located in the lower dimensions.
> 3. In Appendix B, we further show that this dimension heterogeneity originates from RoPE itself. Jamba attention[12], which does not use RoPE, exhibits no such dimension heterogeneity.
>
> Indeed, the reviewer also acknowledges that our empirical finding that different head dimensions serve distinct roles (local vs. global context) is interesting and well-validated through ablation studies, so deeming our work a bit over-claimed appears inconsistent.

---

> ### Author Response · Authors · 2025-11-21
> **Reply (2/5)**
>
> > Regarding W4, W5 and Q4 on different compression rations and ablation study as well as W3 on fair comparison
>
> Thank you for raising this important question. In the case of **Palu**, its paper mainly experiments with three compression ratios—30%, 50%, and 70%—and its codebase also recommends these three ratios as the available options. We therefore adopt the **officially recommended minimum compression ratio of 70%**, which is the closest to our **76%** setting [8]. Although this configuration is slightly more conservative than ours from an algorithmic standpoint, FourierAttention consistently outperforms Palu in both effectiveness and efficiency, demonstrating that we achieve superior cache compression.
>
> When comparing with **SnapKV** and **PyramidKV**, both methods compress the intermediate KV cache to a fixed length by retaining a constant number of tokens [5–6]. We set their number of preserved tokens to match the number of **real-valued frequency states (1024)** that we compress into after transformation (lines 267–268).
>
> Besides, as for your question on different compression ratios and ablation study, we have also considered this aspect and conducted analytical experiments, which are presented in the paper. We experiment with different preserved sizes N' $\in$ {512, 1024, 2048} on Llama 3.2-3B, covering multiple categories of LongBench sub-tasks, including SingleDoc (SD), MultiDoc (MD), FewShot (ICL), and Code. The figure is located in Figure 4 (page 6). The relevant textual description in the paper states that "**We find that, regardless of the value of N', FourierAttention exhibits sufficient robustness and outperforms SnapKV on average across different recall sizes**. We chose N'=1024 in our paper, which is a parameter that balances performance and speed." (page 6 line 317)
>
> Furthermore, as can be observed from Figure 4, the average scores across sub-tasks for N’ $\in$ {512, 1024, 2048} are {42.05, 44.85, 45.24} respectively, indicating that our method's performance on downstream tasks gradually stabilizes when N' is greater than or equal to 1024. These experimental results also demonstrate the robustness of our method. Most importantly, FourierAttention achieves higher average scores across the aforementioned sub-tasks than SnapKV for all values of N'. Therefore, N'=1024 (i.e., N=512) is not an arbitrary choice, but rather a sweet spot that balances the "performance-speed" trade-off for our method.
>
> |                           | SD    |       |       | MD    |       |      | ICL   |       |       | Code  |       | Avg   |
> |:--------------------------|:-----:|------:|:-----:|:-----:|:-----:|:----:|:-----:|:-----:|:-----:|:-----:|:-----:|:-----:|
> |                           | NQ    | Qsp   | MF    | HQ    | WQ    | Msq  | TR    | TQ    | SS    | LCC   | Re-P  |       |
> | FourierAttention(N'=512)  | 9.0   | 21.0  | 34.3  | 10.5  | 13.4  | 6.3  | 69.3  | 86.2  | 37.9  | 59.7  | 59.9  |**42.1**|
> | SnapKV(N'=512)            | 10.6  | 21.0  | 35.0  | 9.5   | 12.7  | 6.6  | 69.5  | 86.3  | 38.3  | 58.1  | 56.9  | 41.3  |
> | FourierAttention(N'=1024) | 12.8  | 21.1  | 35.8  | 9.8   | 11.6  | 6.5  | 69.0  | 87.0  | 39.4  | 69.8  | 63.3  |**44.9**|
> | SnapKV(N'=1024)           | 9.4   | 21.0  | 35.0  | 9.5   | 12.8  | 6.6  | 69.5  | 86.4  | 38.3  | 58.2  | 56.8  | 41.3  |
> | FourierAttention(N'=2048) | 12.2  | 21.2  | 35.7  | 10.5  | 12.8  | 7.2  | 70.0  | 86.9  | 38.2  | 70.2  | 64.3  |**45.2**|
> | SnapKV(N'=2048)           | 9.0   | 21.4  | 35.6  | 9.6   | 12.7  | 6.6  | 69.5  | 85.4  | 38.4  | 58.6  | 57.1  | 41.4  |

---

> ### Author Response · Authors · 2025-11-21
> **Reply (3/5)**
>
> > Regarding W6 and Q3 on evaluation on other LLMs
>
> Thank you for your question. The generalization ability of a method across different model architectures is a key metric for evaluating its effectiveness and practicality. Following your advice, to comprehensively validate the generalizability of our method, we have supplemented our original experiments on the Llama series models with evaluations on Qwen2.5 architecture (the Qwen2.5-1.5B and Qwen2.5-3B models). As with the Llama models, we select the LongBench benchmark for long-context understanding to conduct the evaluation, using an experimental setup similar to that for the Llama models. The experiments show that our method continues to demonstrate strong competitiveness on the Qwen2.5 models. The detailed results are shown in the table below.
>
> |              | SD   |      |      | MD   |      |      | Sum  |      |      | ICL  |      |      | Syn |     | Code |      | Avg. |
> |:-------------|:----:|:----:|:----:|:----:|:----:|:----:|:----:|:----:|:----:|:----:|:----:|:----:|:---:|:---:|:----:|:----:|:-----|
> |              | NQ   | Qsp  | MF   | HQ   | WQ   | Msq  | GR   | QS   | MN   | TR   | TQ   | SS   | PC  | PR  | LCC  | Re-P |      |
> |***Qwen2.5-1.5B***|*10.6*|*24.5*|*36.6*|*23.4*|*21.8*|*11.0*|*30.2*|*23.5*|*26.1*|*71.5*|*77.9*|*42.9*|*2.0*|*7.5*|*56.1*|*58.8*|*36.7*|
> | + SLM        | 12.7 | 13.6 | 26.1 | 10.2 | 14.6 | 5.3  | 21.5 | 20.4 | 23.9 | 56.5 | 56.6 | 40.5 | 1.4 | 5.8 | 48.9 | 46.9 | 28.9 |
> | + SnapKV     | 8.9  | 24.6 | 36.3 | 22.0 | 21.8 | 10.5 | 30.0 | 23.0 | 25.9 | 71.0 | 76.9 | 42.1 | 2.0 | 7.5 | 45.4 | 50.3 | 33.7 |
> | + PyramidKV  | 8.9  | 24.5 | 36.3 | 21.7 | 21.9 | 10.5 | 30.0 | 23.1 | 25.9 | 71.5 | 77.0 | 42.2 | 2.0 | 7.5 | 45.4 | 50.2 | 33.7 |
> | + Palu       | 9.1  | 19.6 | 37.5 | 10.5 | 19.8 | 4.3  | 16.8 | 12.6 | 12.4 | 70.0 | 68.3 | 22.2 | 1.6 | 3.5 | 47.2 | 46.5 | 28.4 |
> | + FA(ours)   | 6.7  | 24.2 | 34.4 | 22.9 | 21.8 | 10.3 | 21.6 | 23.0 | 25.0 | 63.5 | 76.5 | 42.0 | 2.0 | 8.5 | 55.9 | 55.4 |**34.8**|
> |***Qwen2.5-3B***|*14.3*|*29.9*|*45.9*|*38.4*|*33.5*|*20.6*|*29.7*|*24.2*|*25.2*|*69.0*|*89.5*|*45.3*|*2.0*|*8.0*|*69.3*|*65.6*|*42.7*|
> | + SLM        | 13.9 | 17.6 | 25.6 | 21.3 | 23.6 | 7.8  | 22.3 | 19.9 | 24.5 | 51.5 | 79.1 | 42.4 | 1.5 | 6.5 | 56.0 | 52.7 | 33.2 |
> | + SnapKV     | 14.5 | 30.0 | 45.4 | 37.1 | 33.6 | 19.7 | 29.2 | 23.7 | 25.0 | 69.0 | 88.1 | 45.0 | 1.5 | 8.0 | 57.3 | 55.3 | 39.5 |
> | + PyramidKV  | 14.5 | 30.0 | 45.6 | 37.1 | 33.5 | 19.7 | 29.4 | 23.8 | 25.0 | 69.0 | 88.5 | 45.0 | 1.5 | 8.0 | 57.4 | 55.2 | 39.5 |
> | + Palu       | 6.3  | 15.5 | 35.2 | 11.3 | 17.3 | 5.5  | 17.6 | 11.4 | 14.2 | 68.0 | 78.3 | 26.1 | 4.7 | 5.0 | 58.1 | 55.3 | 31.5 |
> | + FA(ours)   | 6.4  | 27.7 | 43.3 | 30.4 | 31.7 | 16.3 | 22.0 | 23.4 | 24.7 | 66.5 | 88.9 | 45.2 | 2.0 | 7.5 | 68.9 | 62.0 |**40.1**|
>
> Compared to various KV Cache compression methods such as SnapKV, our method achieves performance improvements on LongBench, which is consistent with the trend we observed on the Llama models.
>
> Furthermore, to demonstrate the effectiveness of our method with different head dimension sizes (head_dim), we also conduct the same experiments on the Llama 3.2-1B model (which has a head dimension of 64, a significant difference from the 128 of the Llama 3.2-3B model). The results are as follows.
>
> |             | SD  |      |      | MD  |      |     | Sum  |      |      | ICL  |      |      | Syn |     | Code |      | Avg. |
> |:------------|:---:|:----:|:----:|:---:|:----:|:---:|:----:|:----:|:----:|:----:|:----:|:----:|:---:|:---:|:----:|:----:|:----:|
> |             | NQ  | Qsp  | MF   | HQ  | WQ   | Msq | GR   | QS   | MN   | TR   | TQ   | SS   | PC  | PR  | LCC  | Re-P |      |
> |***Llama3.2-1B***|*5.1*|*20.2*|*24.9*|*9.1*|*10.3*|*6.2*|*28.2*|*20.8*|*23.4*|*67.5*|*82.1*|*39.5*|*1.4*|*3.4*|*56.7*|*57.2*|*33.1*|
> | + SLM       | 5.8 | 11.8 | 16.1 | 6.1 | 9.5  | 3.4 | 16.5 | 19.8 | 20.1 | 50.5 | 68.4 | 37.2 | 1.3 | 4.1 | 48.1 | 48.4 | 27.1 |
> | + SnapKV    | 4.1 | 19.7 | 24.2 | 9.3 | 10.1 | 6.0 | 28.4 | 20.4 | 23.2 | 66.5 | 81.2 | 39.2 | 1.5 | 3.4 | 46.2 | 47.7 | 30.2 |
> | + PyramidKV | 4.3 | 19.6 | 24.3 | 9.1 | 10.1 | 6.0 | 28.4 | 20.5 | 23.1 | 67.0 | 81.3 | 39.3 | 1.5 | 3.4 | 46.2 | 47.7 | 30.2 |
> | + Palu      | 1.9 | 10.0 | 13.4 | 3.9 | 7.8  | 1.2 | 10.3 | 4.3  | 9.2  | 51.0 | 28.2 | 12.3 | 0.5 | 3.5 | 44.3 | 40.9 | 19.6 |
> | + FA(ours)  | 2.4 | 17.1 | 24.0 | 8.8 | 10.5 | 4.9 | 16.2 | 19.6 | 19.6 | 63.5 | 80.8 | 39.5 | 1.3 | 4.1 | 56.8 | 51.7 |**30.8**|
>
> The experiments show that our method can achieve consistently effective performance on models with different architectures and head dimensions, which strongly proves that our method possesses good cross-architecture generalization ability. Thank you again for your suggestion. We will organize these detailed experimental results and analysis and incorporate them into the paper.

---

> ### Author Response · Authors · 2025-11-21
> **Reply (4/5)**
>
> > Regarding W7 on the details of dimension selection
>
> Thank you for raising this question. We apologize that the description in the original paper may have caused ambiguity. The following is a clearer explanation of the dimension selection method.
>
> The dimension selection is the result of an **offline** analysis, rather than being performed dynamically and adaptively during inference. The sentence you quoted, "we directly compress and decompress all KV caches, prioritizing dimensions with smaller mean-squared error", describes precisely the core idea behind our offline analysis for establishing the compression principles.
>
> Specifically, we conduct a preliminary study: using the prompt part of the 32k token length NIAH dataset, we perform a one-time forward pass and hook the hidden states of each layer. Then, for each dimension within the caches of different layers and types (K and V), we apply Fourier compression and decompression operations and measure their reconstruction fidelity against the original value (using MSE as the metric). The purpose of this process is to identify which dimensions' information can be more easily captured and reconstructed by our compression algorithm (i.e., the Fourier transform). Dimensions with a lower reconstruction error (MSE) imply that they are inherently "smoother" and, therefore, more suitable for compression.
>
> Thus, the final algorithm does not dynamically select dimensions at runtime. Instead, it applies a fixed policy based on this offline dataset analysis and an "asymmetric, inverted pyramid" compression ratio scheme (i.e., K is compressed more, and earlier layers are compressed more).
>
> We hope this clarification resolves your question. Thank you for your valuable feedback. We will articulate this point more clearly in the revised version of the paper to prevent similar confusion for future readers.

---

> ### Author Response · Authors · 2025-11-21
> **Reply (5/5)**
>
> > Regarding Q2 on the numerical results of efficiency comparison
>
> Thank you for raising this good question. Below are the detailed experimental results for prefill latency and reserved memory from our paper. We have also supplemented experiments on throughput during the decoding process for each model at various context lengths, which are presented below. As shown in the table, we have achieved a latency close to that of existing efficient approaches such as Palu and SnapKV in long contexts ranging from 16k to 80k, while the memory footprint is notably lower than other methods across all context lengths. In addition, our method demonstrates higher throughput than other approaches during the decoding phase. We will carefully revise this part in our paper. Thank you for your review.
>
> | Peak Memory (MB) | 8K     | 16K    | 32K    | 48K    | 64K    | 80k    |
> |:-----------------|:------:|:------:|:------:|:------:|:------:|:------:|
> |***Llama 3.1-8B***|        |        |        |        |        |        |
> | + SnapKV         | 26.27  | 37.44  | 59.78  | 77.29  | oom    | oom    |
> | + Palu           | 24.56  | 32.56  | 47.31  | 62.49  | 77.65  | oom    |
> | + FA (ours)      | 18.37  | 21.39  | 27.42  | 33.51  | 39.61  | 45.72  |
> |***Llama 3.2-3B***|        |        |        |        |        |        |
> | + SnapKV         | 14.69  | 26.15  | 40.46  | 57.65  | oom    | oom    |
> | + Palu           | 14.85  | 21.92  | 36.05  | 49.91  | 63.95  | 71.46  |
> | + FA (ours)      | 9.20   | 12.08  | 17.03  | 23.57  | 29.36  | 35.15  |
>
> | Prefill Latency (s) | 8K    | 16K   | 32K   | 48K   | 64K   | 80k   |
> |:--------------------|:-----:|:-----:|:-----:|:-----:|:-----:|:-----:|
> |***Llama 3.1-8B***   |       |       |       |       |       |       |
> | + SnapKV            | 0.29  | 0.66  | 1.65  | 3.06  | oom   | oom   |
> | + Palu              | 0.31  | 0.69  | 1.81  | 3.33  | 5.23  | oom   |
> | + FA (ours)         | 0.37  | 0.76  | 1.80  | 3.30  | 5.04  | 7.29  |
> |***Llama 3.2-3B***   |       |       |       |       |       |       |
> | + SnapKV            | 0.15  | 0.36  | 0.93  | 1.75  | oom   | oom   |
> | + Palu              | 0.16  | 0.38  | 0.99  | 1.88  | 3.00  | 4.32  |
> | + FA (ours)         | 0.21  | 0.44  | 1.03  | 1.95  | 2.99  | 4.44  |
>
> | Throughput (token/s) | 8K      | 16K    | 32K    | 48K    | 64K    | 80k    |
> |:---------------------|:-------:|:------:|:------:|:------:|:------:|:------:|
> |***Llama 3.1-8B***    |         |        |        |        |        |        |
> | + SnapKV             | 77.49   | 40.35  | 19.73  | 16.50  | oom    | oom    |
> | + Palu               | 109.54  | 50.91  | 27.61  | 17.32  | 13.66  | oom    |
> | + FA (ours)          | 127.05  | 64.58  | 35.02  | 21.23  | 17.46  | 11.81  |
> |***Llama 3.2-3B***    |         |        |        |        |        |        |
> | + SnapKV             | 118.32  | 60.85  | 30.48  | 23.56  | oom    | oom    |
> | + Palu               | 152.75  | 70.73  | 39.35  | 29.96  | 18.65  | 14.88  |
> | + FA (ours)          | 185.20  | 93.14  | 55.25  | 33.67  | 24.16  | 20.72  |
>
> [1] KV Cache Compression, But What Must We Give in Return? A Comprehensive Benchmark of Long Context Capable Approaches https://arxiv.org/abs/2407.01527
>
> [2] Challenges in Deploying Long-Context Transformers: A Theoretical Peak Performance Analysis https://arxiv.org/abs/2405.08944
>
> [3] A Survey on Large Language Model Acceleration based on KV Cache Management https://arxiv.org/abs/2412.19442
>
> [4] Efficient Streaming Language Models with Attention Sinks https://arxiv.org/abs/2309.17453
>
> [5] SnapKV: LLM Knows What You are Looking for Before Generation https://arxiv.org/abs/2404.14469
>
> [6] PyramidKV: Dynamic KV Cache Compression based on Pyramidal Information Funneling https://arxiv.org/abs/2406.02069
>
> [7] Model Tells You What to Discard: Adaptive KV Cache Compression for LLMs https://arxiv.org/abs/2310.01801
>
> [8] Palu: Compressing KV-Cache with Low-Rank Projection https://arxiv.org/abs/2407.21118
>
> [9] Massive Values in Self-Attention Modules are the Key to Contextual Knowledge Understanding https://arxiv.org/abs/2502.01563
>
> [10] VideoRoPE: What Makes for Good Video Rotary Position Embedding? https://arxiv.org/abs/2502.05173
>
> [11] Scaling Laws of RoPE-based Extrapolation https://arxiv.org/abs/2310.05209
>
> [12] Jamba-1.5: Hybrid Transformer-Mamba Models at Scale https://arxiv.org/abs/2408.12570

---

> ### Author Response · Authors · 2025-11-25
> **Looking forward to receiving your feedback**
>
> Dear Reviewer smj6,
>
> We hope this message finds you well. We truly appreciate the time and effort you’ve dedicated to reviewing my work, and we would be very grateful if you could provide feedback on our rebuttal. If you require further clarification or have any additional concerns, please do not hesitate to contact us. We are more than willing to continue communicating with you.
>
> Best wishes,
>
> The Authors

---

> > ### Comment · Reviewer_smj6 · 2025-11-27
> >
> > I have carefully read the rebuttal and the new experiments. I appreciate the authors' effort in answering my questions. While some of the answers have confirmed my assessment concerning fair compression ratios comparison and method presentation, I acknowledge the new ablations and experiments, and I will raise my score to 4.

---

### Comment · Area_Chair_F82R · 2025-11-27

Dear reviewers,

A reminder that the discussion phase will end in a few days (**December 2**). Engaging with the author's rebuttal is essential to address all potential concerns before our final discussion stage.

Thanks,
The AC

---

### Author Response · Authors · 2025-12-02
**Summary for AC (1/2)**

Dear AC,

We appreciate your time and effort in reviewing our submission, especially considering the recent challenges. We thank you for your attention to our work.

We also thank the reviewers for their valuable and constructive review. We are encouraged that reviewers acknowledge the contribution of our work in providing
- a novel and well-validated observation as well as analysis on dimension heterogeneity (wpka, iCAS, sDbS),
- a mathematically rigorous compression scheme with a distinct perspective (wpka, smj6),
- a thorough comparison in terms of performance and latency, effectively demonstrating the superiority over other competitive cache optimization baselines (iCAS, sDbS), and
- the solid system implementation based on the custom Triton kernel (wpka, iCAS, sDbS, smj6).

The reviewers also raise a number of questions, which we respond to in the reviewer-specific comments below. Altogether, our responses provide further support for our paper's central claim that *we propose FourierAttention, a training-free memory-efficient framework that exploits the heterogeneous roles of transformer head dimensions*: lower dimensions prioritize local context, while upper ones capture long-range dependencies.

**Regarding Model Generalization** (wpka, iCAS, sDbS, smj6)

This is an important question raised by all four reviewers. We have added results for other model families, Qwen2.5-1.5B and Qwen2.5-3B, as well as for Llama3.2-1B with a different head dimension, and have run all the methods compared previously on Llama3.1-8B and Llama3.2-3B. FourierAttention consistently outperforms all baselines across these models. Concerning statistical significance (sDbS), our method yields small but consistent advantages, delivering the closest scores to the original model while achieving the best memory efficiency. The new experimental results will be included in the revised version.

**Regarding Efficiency Evaluation** (wpka, sDbS, smj6)

We report peak memory and pre-filling latency vs. SnapKV and Palu as bar and line charts in the original paper, and we provide throughput comparisons alongside complete numerical results in the rebuttal. Our FourierAttention achieves the lowest memory cost and the highest throughput, enabling longer context on a single A100 than SnapKV and Palu, and confirming its memory efficiency. Concerning the OOM issue raised by Reviewer wpka, we clarify that SnapKV’s OOM stems from the prohibitive memory and latency overheads incurred by attention-score recalculation in the pre-filling phase, as we state in the Introduction. **Reviewer wpka has also confirmed the fairness of the comparison**. All numerical results will be added to the revised version.

---

### Author Response · Authors · 2025-12-02
**Summary for AC (2/2)**

**Regarding Dimension Selection** (sDbS, smj6)

We first reply to Reviewer smj6 with a refined clarification of our dimension selection. Regarding the sensitivity raised by Reviewer sDbS, we have already presented several comparative experiments that indirectly validate the effectiveness of our dimension selection scheme. For example, in Table 2, for the inverted-pyramid, asymmetric compression scheme mentioned in the paper, we test the following alternative strategies on the Llama3.2-3B model for the ruler 4k under an equivalent compression ratio. Furthermore, we also conduct a new experiment to simulate small-scale, random errors in dimension identification by randomly replacing 10% of these long-context-sensitive dimensions with an equal number of randomly selected long-context-insensitive dimensions. The results show that with this 10% dimension identification error rate, the model's final performance degrades severely, confirming that only a few dimensions are critical for long-context tasks and that compressing the rest to a fixed length is justified.

**Regarding Dimension Heterogeneity** (wpka)

In our reply to Reviewer wpka, we first show that the long-context-sensitive heads labeled by the existing head-heterogeneity methods, DuoAttention and HeadKV, are highly consistent with the heads labeled with more long-context-sensitive dimensions by our FourierAttention. If we instead keep only those heads uncompressed and compress the rest to a fixed length, long-context performance drops sharply. These two facts demonstrate the fundamental reason why an attention head becomes important for the long context is that it contains more dimensions responsible for processing the long-context information.

Regarding Reviewer wpka’s feedback on the number of upper or lower dimensions, we adopt the critical dimension as the boundary. Our experiments show that the two groups behave very differently on long-context tasks. The ratio of the critical dimension over the head dimension is determined by the pre-training context length and rotary base, yielding ~60% for Llama and Qwen, thus being model-adaptive. **These replies have addressed Reviewer wpka's concern**.

**Regarding Ablation Study** (smj6)

We already present an ablation study on compression ratios in Figure 4 and supply the corresponding numerical results in tabular form in the rebuttal. Other reviewers have confirmed these comparisons are thorough and fair. We find that, regardless of the value of N', FourierAttention exhibits sufficient robustness and outperforms SnapKV on average across different recall sizes, confirming the effectiveness of compression based on HiPPO-FourierT. **Reviewer smj6 has appreciated our effort and acknowledged the new ablations and experiments**.

**Regarding Method Presentation** (smj6)

Our novelty has been acknowledged by other reviewers, and we have further clarified to Reviewer smj6 the unique contributions of our work. Earlier studies only note that certain dimensions are more important in long contexts, but these observations remain case-level or speculative. They neither demonstrate the role of dimensional heterogeneity in attention scores, long-context downstream tasks, and cache optimization, nor provide an in-depth theoretical framework, as we have done in our paper.

---

### Meta-Review · Area_Chair_j6vD · 2026-01-13

**Summary:**

Reviewers generally agree that the paper presents a technically sound and thoughtfully motivated approach to KV-cache compression based on heterogeneous long-context sensitivity across attention dimensions. The idea of selectively applying Fourier-based approximation to “long-context-insensitive” dimensions is viewed as novel in formulation and conceptually interesting. However, the dominant concern across reviews is that the empirical gains over strong and closely related baselines such as SnapKV and PyramidKV are modest and sometimes mixed, making it difficult to justify acceptance on impact alone in a highly competitive area. Several reviewers also expressed reservations about the clarity and completeness of the efficiency evaluation, including whether the reported improvements are consistently meaningful across models and settings. Taken together, even after rebuttal, these concerns lead to the conclusion that the paper remains close to but below the acceptance bar, which directly motivates the recommendation to reject.

**Reviewer Concerns:**

The rebuttal substantively addressed a number of concrete reviewer requests. The authors added experiments on additional model families beyond LLaMA-3 (including Qwen2.5 and smaller LLaMA variants), clarified situations in which baseline methods encounter OOM behavior, and expanded discussion of efficiency metrics such as throughput and prefill costs. These changes strengthen the paper and improve confidence in correctness and generalization. However, the rebuttal does not fundamentally alter the central concern about impact: the improvements remain relatively small relative to strong baselines, and reviewer skepticism about whether these margins are sufficient to justify publication is only partially alleviated. As a result, while the rebuttal improves clarity and scope, it does not shift the overall balance of opinion toward clear acceptance.

**Reviewer Scores:**

Reviewer iCAS gave an initial score of 8 and expressed strong support for the paper. There is no indication that this reviewer participated further in the discussion, and nothing in the rebuttal would plausibly reduce their confidence. The most reasonable expectation is that this reviewer remains at 8.

Reviewer wpka gave an initial score of 4. In the discussion, this reviewer acknowledged that the rebuttal and added experiments addressed several of their earlier concerns, particularly regarding generalization beyond a single model family and clarification of efficiency-related issues, and explicitly stated that they would raise their score to 6. While this reflects increased confidence after rebuttal, the revised score still indicates a cautious stance rather than strong endorsement.

Reviewer sDbS gave an initial score of 4 and did not indicate a change in stance during the discussion. Their core concern about limited empirical margins and unclear statistical robustness is only partially mitigated by the rebuttal, and the most reasonable expectation is that this reviewer remains at 4.

Reviewer smj6 participated actively in the discussion and explicitly stated that, after reading the rebuttal and new experiments, they would raise their score to 4. This explicit signal indicates some improvement in confidence, but the reviewer remained below the acceptance threshold.

Overall, after rebuttal, the score distribution is characterized by one strong accept (8), one cautious weak accept (6), and two below-threshold evaluations (4, 4), which does not provide sufficient consensus for acceptance under a selective quota.

---

### Decision · Program_Chairs · 2026-01-26

Reject